# Ship wake-induced water column mixing and meter-scale seabed erosion in the Baltic Sea

Jacob Geersen [1] ✉, Peter Feldens [1], Luisa Rollwage [2,3],
Lenya Mara Baumann [2], Knut Krämer [2], Patrick Westfeld [4],
Sebastian Krastel [2], Soeren Ahmerkamp [1], Franz Tauber[1] &
Jens Schneider von Deimling [2]

Commercial shipping is a cornerstone of global trade. Its impact on the marine environment, however, remains underexplored. This study combines hydro-acoustic data, sediment samples, propeller-induced shear stress calculations and vessel tracking information to assess the effects of shipping in one of the busiest maritime regions in the Baltic Sea, the Bay of Kiel. We unveil substantial seafloor erosion, including up to 1.5 m variation in water depths, over 10 years that clearly relates to vessel traffic. By imaging water column disturbance behind passing ships, we trace wake turbulence to the seafloor and show the breakdown of a strongly stratified water column and a possible excitement of internal waves, likely increasing the mixing of oxygen, nutrients, and green-house gases. While the environmental consequences of this anthropogenic stressor are unquantified, our findings leave little doubt that they include modifications to marine ecosystems and element budgets on a Baltic-wide scale.

The Baltic Sea is a shallow European semi-enclosed marginal sea that is strongly affected by climate change and human activities[1–3]. It covers an area of 415,000 km², and its catchment basin, which is four times larger, is home to approximately 85 million people. Ever-increasing conflicts of interest arise from the rising demand of multiple socio-economic players such as offshore wind, nature conservation, shipping, coastal protection, fishing, military, tourism and many more[1]. About 26% of the Baltic Sea is shallower than 20 m. In the southwestern Baltic Sea (west of 15°E), the regions shallower than 20 m even account for 56% of the entire area (Fig. 1a). Consequently, many of the anthropogenic pressures at work affect the entire water column, from the ocean-atmosphere boundary down to the seafloor and below.

One of the most widespread anthropogenic stressors affecting life below water is commercial shipping and marine trade[4] (Fig. 1b). The consequences of commercial shipping are, however, only partly researched. This is partly because the water column and the seafloor are veiled from our eyes so that we rely on geophysical methods to resolve their integrity and to trace anthropogenic impacts[5]. Increased marine traffic over the last decades has already led to an increased number of bigger ships, more powerful propulsion systems, and increased ship noise[6]. The morphological impact of ships anchoring on the seafloor has been studied by different authors[7,8]. Ship bubble wakes, and their detection near the sea surface by optical and acoustic sensing, have also been widely investigated[9,10]. Nylund et al. suggested[11] that passing ships can trigger methane emissions from natural sources in coastal areas. For confined areas like ports, Guarnieri et al. discussed[12] the influence of ship propellers on sediment erosion and accumulation. What has, however, only been marginally researched is the impact of the wake of passing ships on the water column, sedimentation patterns, seafloor morphology and the benthic ecosystem in open waters[13–15]. For the Venice lagoon, Madricardo et al. and Scarpa et al. showed[16,17] that depression wakes created by large ships cause the shoreline to retreat at different locations, thereby threatening the stability of anthropogenic structures.

[1]Leibniz Institute for Baltic Sea Research Warnemünde (IOW), Rostock, Germany. [2]Institute of Geosciences, Kiel University, Kiel, Germany. [3]Now at: School of Earth and Environment, University of Canterbury, Christchurch, New Zealand. [4]Federal Maritime and Hydrographic Agency of Germany (BSH), Rostock, Germany. ✉e-mail: jacob.geersen@iow.de

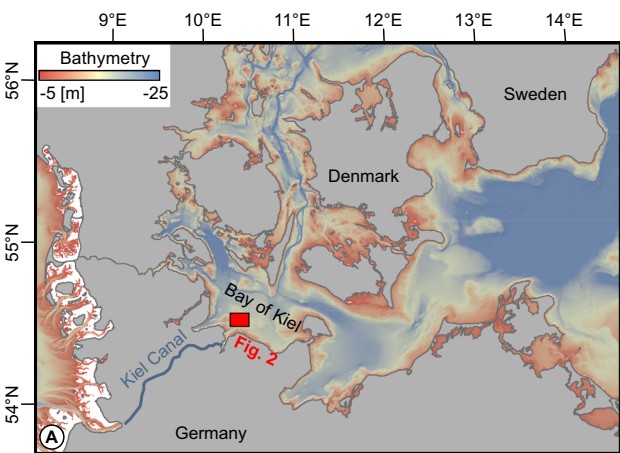

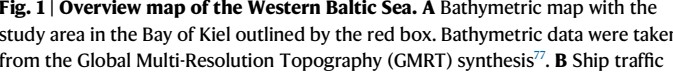

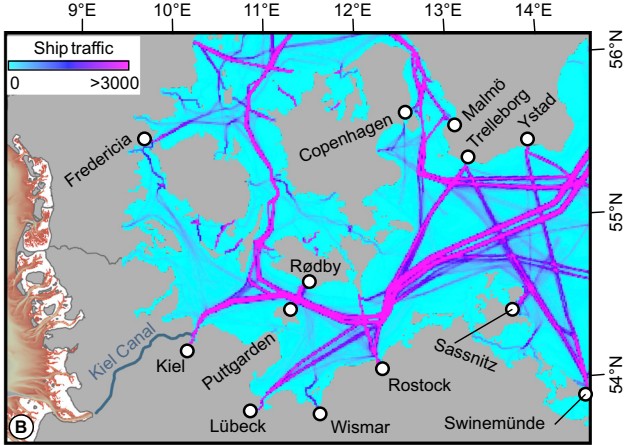

**Fig. 1 | Overview map of the Western Baltic Sea. A** Bathymetric map with the study area in the Bay of Kiel outlined by the red box. Bathymetric data were taken from the Global Multi-Resolution Topography (GMRT) synthesis[77]. **B** Ship traffic intensity[2] for the year 2022 in the Western Baltic Sea, defined as the number of ships crossing a 1 × 1 km grid cell (max intensity = 37129).

To investigate the impact of commercial shipping on the marine environment, we focus on the Bay of Kiel (Fig. 1). The study area has bathymetry shallower than 20 m and is an area of intense commercial shipping activity due to the entrance to the Kiel Canal (Fig. 1), one of the most heavily used artificial waterways on the globe with around 90 passages per day on average. It is further affected by daily ferry connections to Sweden and Norway, as well as weekly connections to Lithuania. Marine traffic in and out of Kiel and Kiel Canal is guided through a narrow traffic separation scheme in the Bay of Kiel (Fig. 2). In a recent publication, Díaz-Mendoza et al. report[18], that the Bay of Kiel hosts various bedforms that may relate to propeller-induced scouring caused by ships. They documented subaqueous dunes and scouring features in water depths between 10–19 m. Krämer et al. further discussed[15] the small-scale morphological characteristics of these features and showed that their genesis is controlled by changes in bottom shear stress induced by the propeller wakes from passing ships. The Bay of Kiel was mainly shaped during the Weichselian glaciation[19]. Its shallow geologic strata are characterized by basal till that was left behind by the retreating glaciers. In the southwestern Baltic Sea region, till consists of a mixture of siliciclastic material of varying grain size, ranging from clay to boulders, and finely dispersed calcium carbonate[20]. In addition to being present at the seafloor, till also forms the dominant surface lithology along the southwestern Baltic coast. In some central parts of the Bay of Kiel, the till is covered by organic-rich Holocene sediments[21]. Some of these sediments, including mobile sand layers, result from till-abrasion at the seafloor or erosion of coastal cliffs[22–24].

In this work, we demonstrate that ship wakes not only generate distinct morphological features at the seafloor but also completely alter the internal structure of the water column by mixing stratified waters of different oxygen, temperature, and salinity content. With repeated bathymetric surveys, we further show that absolute changes in water depth caused by the erosion and redistribution of sedimentary strata locally exceed 10% on annual to decadal timescales. The implications of these findings for benthic ecosystems on a Baltic-wide scale require dedicated monitoring strategies.

## Results

All hydroacoustic and geological data are located in the central Bay of Kiel (Fig. 1). The bulk of the multibeam echosounder data was recorded in 2014 (Fig. 2) by the German Federal Maritime and Hydrographic Agency (BSH). RV Alkor cruise AL619 in 2024, re-surveyed an area of 5.4 km² located within the previously surveyed region[25] (Fig. 3a). The surveys in 2014 and 2024 allow us to quantify absolute changes in seafloor depth between 2014 – 2024, and to investigate morphological changes (Fig. 4). During the AL619 cruise we also collected six grab samples from which we used the upper 2 cm for grain size analysis (inset Fig. 4a). To link seafloor morphology and ship traffic intensity, we systematically analyzed Automatic Identification System (AIS) data from the study area covering 116 days (1.6.–24.9.2024) prior to the recording of the 2024 multibeam bathymetric data (Fig. 5). To capture the seafloor modulating effect of shipping, we traced the bubbly wake of three commercial ships and two ferries, using a fishery split-beam echosounder optimized for water column analyses (Fig. 6A). The echogram patterns were interpreted together with density variations in the water column, derived from five conductivity, temperature, and depth (CTD) profiles (Fig. 6B). To link our observations to the physical processes of seabed erosion, we calculated seabed shear stress (Fig. 7) induced by the propeller jets of the five ships that we traced with the fishery split- beam echosounder. The stresses are compared to the critical shear stresses required to initiate bedload and suspended load transport of sand.

### Description of morphological units

Seafloor morphology varies systematically in the study area and was subdivided into three morphological units (Fig. 2). Unit 1 is in the southern part of the studied area at around 15–20 m water depth. The seafloor within unit 1 appears flat and featureless and thus hosts an overall low roughness (Fig. 2b). The only prominent structure within unit 1 is an about 2.5 km long meandering ridge that is elevated up to 2 m above the surrounding seabed. In other regions of the Baltic Sea, morphologically similar structures have been interpreted as eskers[26] (Fig. 2a). The boundary between unit 1 and the other units is gradual. Its overall flat character and low roughness indicate that unit 1 corresponds to muddy Holocene sediments that were deposited after the last glaciation[27,28].

Unit 2 occurs along the northwestern and eastern margins of the studied area in water depths between 10 and 14 m (Fig. 2). The seafloor of unit 2 appears hummocky in comparison to unit 1. Regionally, the smooth seafloor forms longer-wavelength undulations that are typical for basal till surfaces that crop out at the seafloor.

The generally hummocky seafloor of unit 2 is crossed by an about 3 km wide, southwest-striking band of higher seafloor roughness (unit 3) (Fig. 2b). Seafloor morphologic structures within unit 3 include:
- Hundreds of depressions with depths of up to 1 m below the surrounding seafloor that usually host objects >20 cm high, most likely boulders. Often, individual depressions merge into larger depressions of up to a few thousand square meters (Fig. 4b, c).

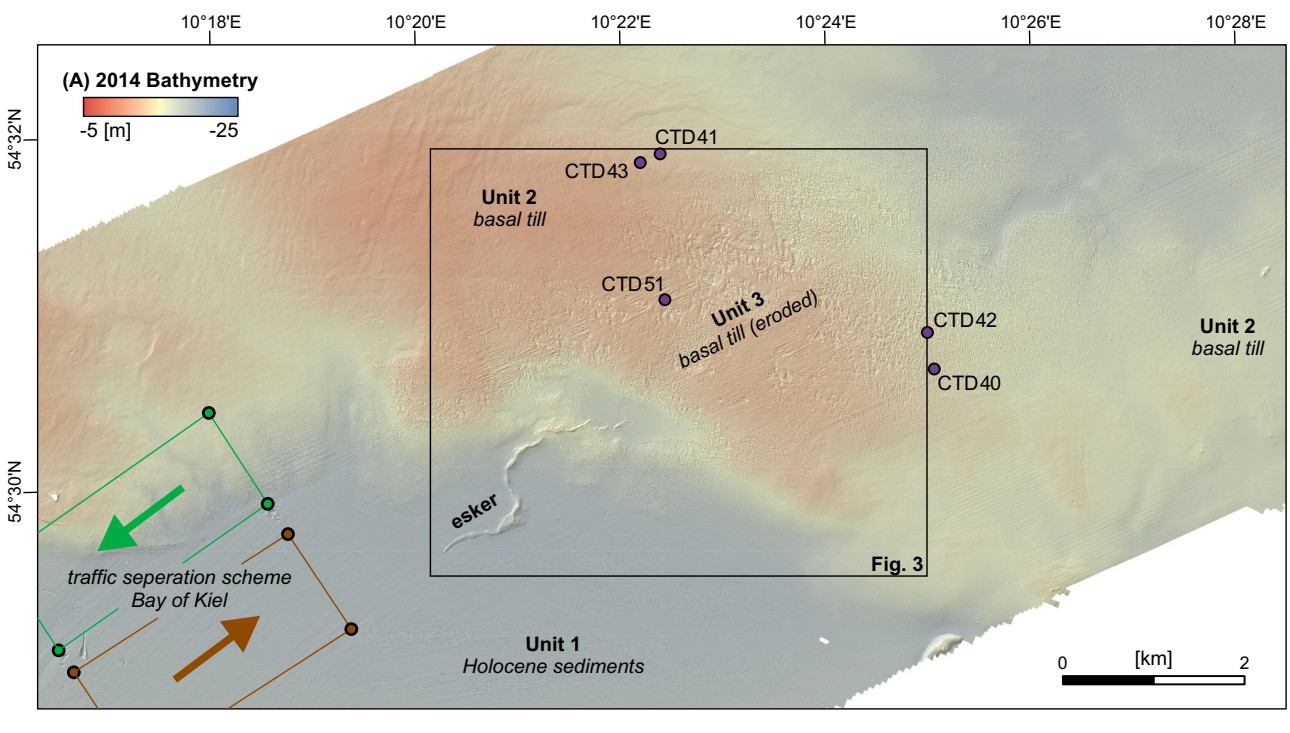

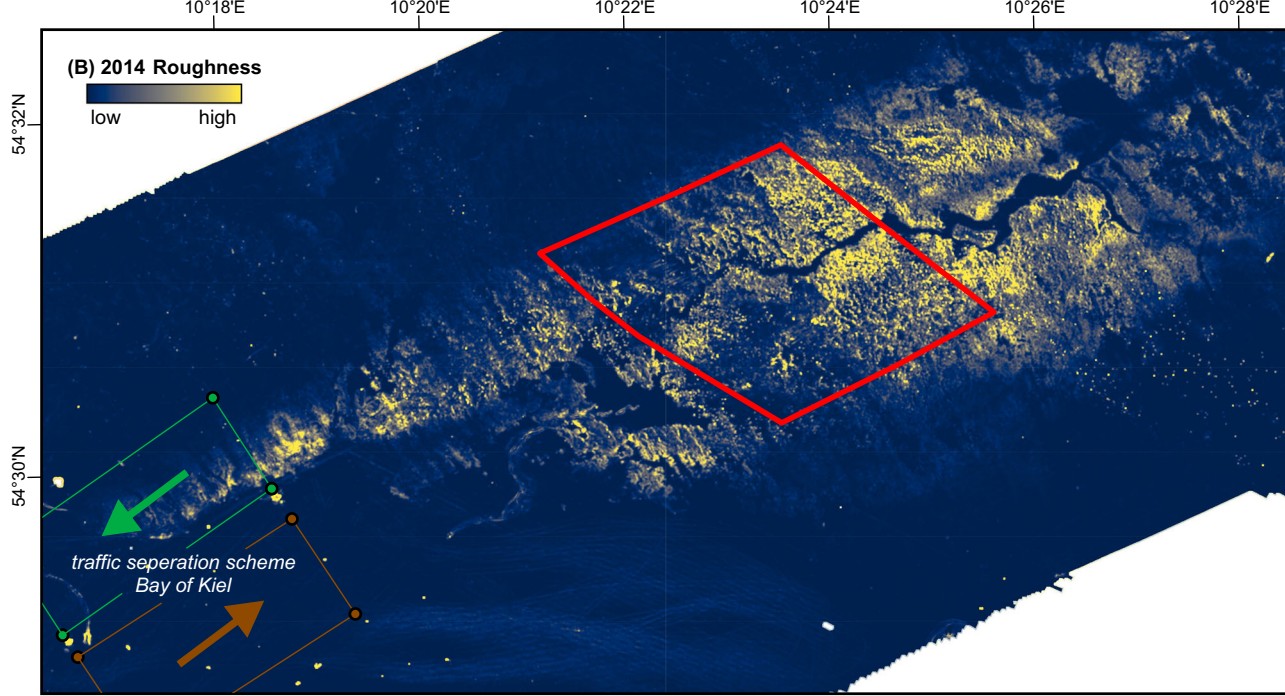

**Fig. 2 | Detailed seafloor morphology in the study area in the Bay of Kiel.** The map is based on bathymetric data collected in 2014 by the German Federal Maritime and Hydrographic Agency (BSH), which has been gridded to a resolution of 1 * 1 m. **A** Bathymetric map of the study area. The traffic separation scheme guides ships towards and out of Kiel, including all traffic that goes through the Kiel Canal (green and brown arrows, respectively). Purple dots are conductivity, temperature, and depth (CTD) stations shown in Fig. 6B. **B** Seafloor roughness derived from the bathymetric data, which spatially coincides with the main shipping corridor. Red box marks the area for which the volume calculations for the depressions were conducted.

- Northeast–southwest extending trains of sand dunes up to 20 cm high with crestlines striking in northwest–southeast direction (Fig. 4b)
- Two linear, northeast–southwest trending ridges with a width of about 30 m, elevated up to 1.5 m above the surrounding seafloor, and that run almost parallel over >5 km (Figs. 3–4). The ridges are most pronounced in the 2024 data and lie oblique to

survey directions, hence are not a result of survey artifacts (Fig. 3a).
- Smooth seafloor patches, which are located adjacent to many of the depressions (Fig. 4b, c).

The grain size measurements conducted on six grab samples that were taken at different positions within unit 3 (AL619-55–AL619-60,

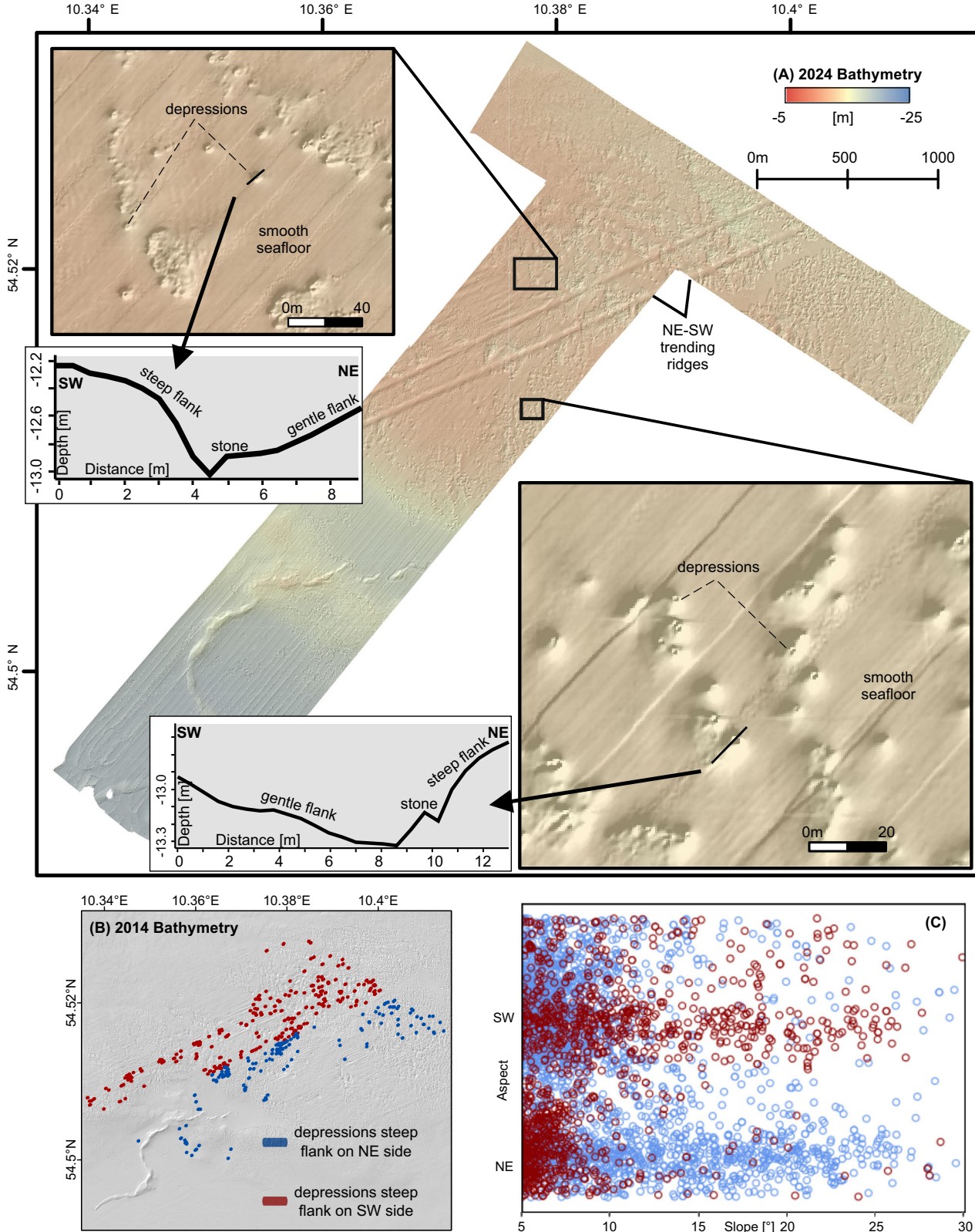

**Fig. 3 | Detailed seafloor morphology in a subset of the study area based on bathymetric data collected in 2024 by the Geophysics and Hydroacoustic Group of Kiel University.** The data has been gridded to a resolution of 0.5 * 0.5 m. **A** Bathymetric map of a subsection of the study area (see Fig. 2 for location). Inset maps show the detailed morphology of the depressions within unit 3. The large stones located within most depressions are well resolved. Note that the survey was oriented in a northeast–southwest direction and that the lines in the inset figures are artefacts of the data. The inset depth profiles are located along the long axis of two selected depressions from the northwestern and southeastern study area. The profiles highlight the reversed orientation of the steep and the gentle dipping flanks. **B** Spatial distribution of depressions that have a steep flank on their NE side (blue) as well as those that host the step flank on their SW side (red). **C** Cross plot of slope versus dip direction for the depressions shown in (**B**) extracted along depth profiles trending similar to those shown in (**A**).

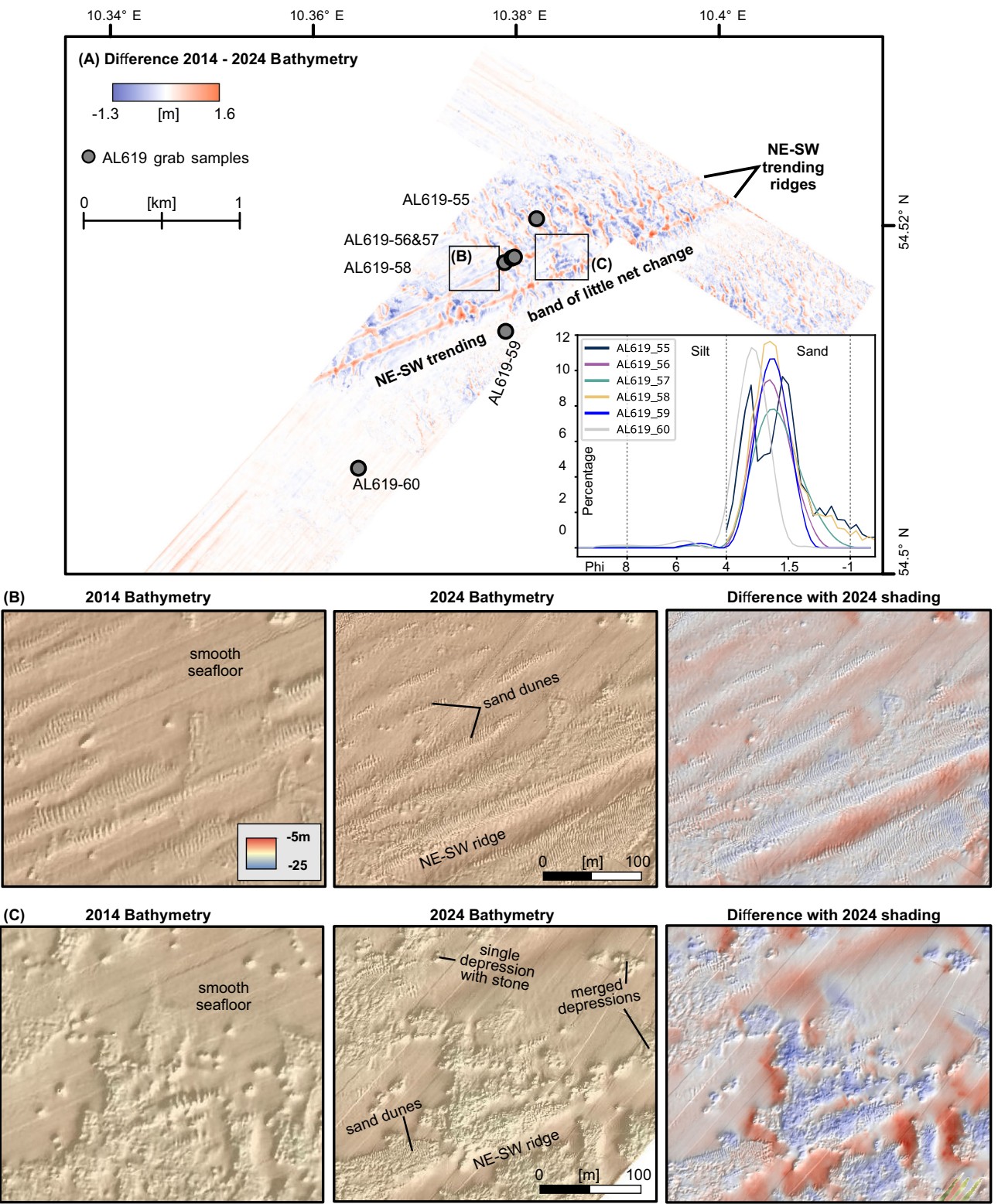

**Fig. 4 | Bathymetric changes between the 2014 and 2024 surveys. A** Main changes occur within the northwestern and eastern study area, with a band of little net change in between. Inset diagram shows the grain size distribution for six grab samples (gray dots) taken during RV ALKOR Cruise AL619. In the samples, which are solely located within unit 3, the fine sand fraction dominates. **B**, **C** Selection of data showing how two areas changed bathymetry between 2014 and 2024.

Fig. 4a), show that the very shallow sedimentary strata of unit 3 consist of sand. In all six samples, the fine to medium sand fraction dominates. Similar to unit 2, we interpret unit 3 as outcropping basal till that has, however, been subject to extensive localized meter-scale erosion. The boulders, gravel, and sand fraction of the eroded till remain within the area, whereas the fine fraction is transported away.

Using a morphological workflow optimized for mapping of seabed depressions[29], we semi-automatically mapped the outlines of the depressions in a 7.2 km² large area in Unit 3 (red outline in Fig. 2b) based on the 2014 bathymetric data. Their cumulative volume is 451,587 m³, which represents 0.063 m³/m² over the whole investigated area. Individual depressions are around 10 m wide and up to 1 m deep

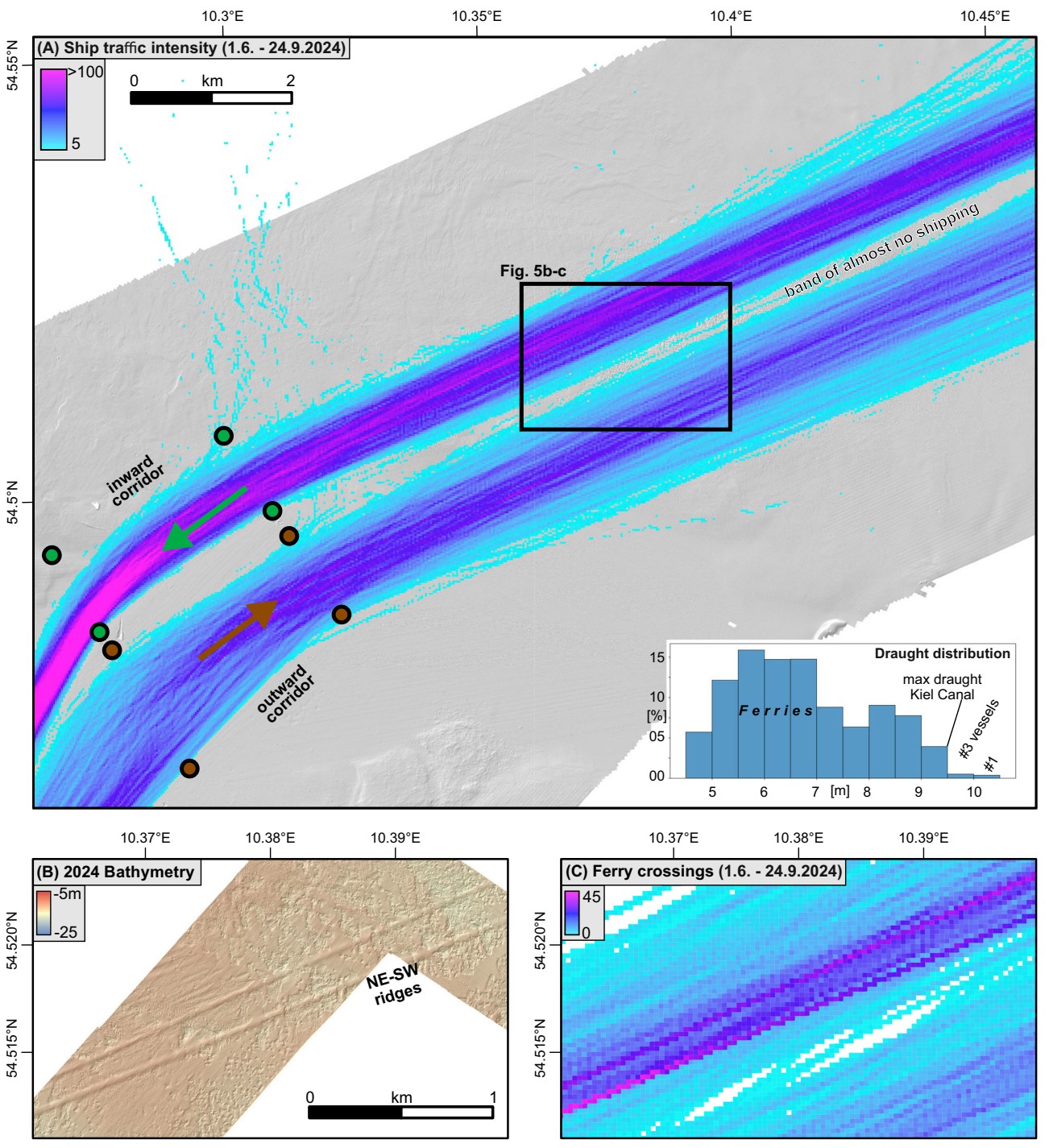

**Fig. 5 | Ship traffic intensity in relation to seafloor morphology. A** Ship traffic intensity in the study is between 1 June–24 September 2024, considering only ships with a draught >4.5 m and a speed >2 kn. Shown is the number of crossings for 25*25 m-sized grid cells (max intensity = 164). Inset diagram shows the draught distribution for the considered ships. **B** Detailed bathymetry in the area of the two linear ridges that were observed in the 2024 bathymetric data. **C** Course of the six ferries that link Kiel with Sweden, Norway and Lithuania (same area covered by Fig. 5B).

(Fig. 3a). They are predominantly elliptical in plan-view with their longest axis striking northeast–southwest. Along their longest axis, the depressions have asymmetric profiles, with a steep (up to 30°) flank on one side and a gently sloping flank on the opposite side (insets Fig. 3a). The orientation of their longest axes with the bimodal pattern of a steep and a gently dipping flank is uniform throughout the study area. The orientation of both flanks, however, varies across the study area. In the northwestern area of unit 3, the steepest flank is on the southwestern side of the depressions (red circles, Fig. 3b, c). This tendency reverses in the southeastern area of unit 3. There, the steep flank forms the northeastern side of the depressions (blue circles, Fig. 3b, c). Individual boulders in the depressions are located close to the above-mentioned steep flanks of the depressions. Depressions without visible boulders may result from the limited horizontal resolution due to the sonar beam footprint which hinders detection of objects smaller than 20 cm at the given water depth.

## Repeat bathymetric survey
In September 2024, we re-surveyed a 5.4 km² area located within the 2014 dataset during R/V Alkor cruise AL619[25] (Fig. 3a). The new data

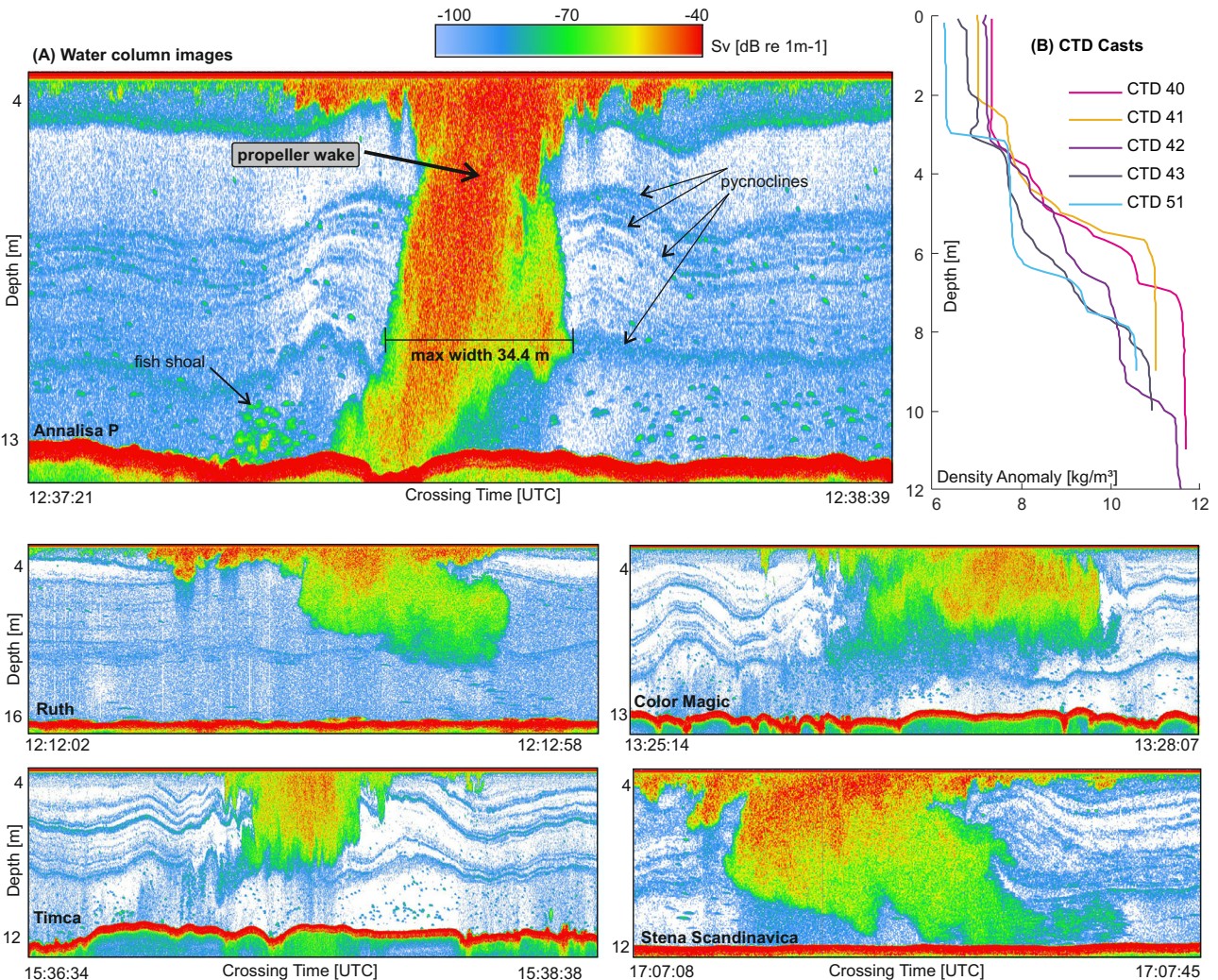

**Fig. 6 | Structure and disturbance of the water column from acoustic volume backscattering (Sv) and conductivity, temperature, and depth (CTD) data. A** Ship wake induced water column disturbance, recorded with an EK60 fishery split-beam echosounder at 120 kHz. The wake corresponds to the 5 ships listed in Tab. 1. It was imaged between 176 m (Annalisa P) – 416 m (Timca) behind the passing ships. **B** Density anomaly of the water column from five CTD casts conducted within the study area on the same day when the EK60 data were recorded (see Fig. 2a for locations of CTD sites).

overlies the morphologically most complex unit 3. It confirms the existence and morphological characteristics of the above-described structural elements within unit 3. The 2024 data reveal northeast–southwest trending ridges (Fig. 3a) that were less prominent in the 2014 data. The distribution of the individual morphologic features has changed over the 10 years, inducing absolute vertical changes of up to 1.5 m (Fig. 4). The largest changes are observed within or at the rim of the large depressions and along the two linear ridges. Here, the shallow sands have been systematically redistributed within the area.

Figure 4a shows that the changes in water depth between 2014–2024 do not occur equally within the re-surveyed area but concentrate within two areas separated by a band of little net change. One area is within the inward corridor of shipping (see paragraph on AIS data below), where the northeast–southwest trending ridges are concentrated. The second area matches the outward corridor of shipping in the eastern part of the study area. The band of little net change between the two areas corresponds to the band of almost no shipping (Fig. 5).

### Automatic identification system (AIS) data
Shipping within the study area is concentrated along two parallel lanes along the Kiel-Baltic Route that trend in northeast–southwest direction

(Fig. 5a). Each strip is around 1 km wide and runs into the inward or outward corridor of the traffic separation scheme. In between the two lanes, a 1 km wide band is rarely crossed by ships. Over the 116 days between 1.6.– 24.9.2024, 1313 ships with a draught >4.5 m and a sailing speed of >2 kn returned their locations through the AIS in the study area. Many of these ships, including the ferries, have passed multiple times through the study area. The cumulative number of passings amounts to 5648, representing 49 passages per day on average. The draught of the ships peaks between 5.5–7 m and 8–9 m (inset Fig. 5a). The first peak relates to the draught of the six ferries (5.5–7.0 m) that connect Kiel with Oslo, Gothenburg and Klaipeda on a daily to sub-weekly basis. The maximum draught (10 m) corresponds nearly with the maximum draught allowed in the Kiel Canal (9.5 m). Only four vessels with drafts >10 m sailed through the area, likely corresponding to berthings at the Port of Kiel, which currently accommodates ships with a maximum draught of up to 11.5 m.

### Water column structure from acoustic backscattering and CTD data
To investigate the extent of a ship wake in the water column, we slowly passed through the wake of three cargo ships and two ferries, using an EK60 fishery split-beam echosounder (Fig. 6). The echograms were

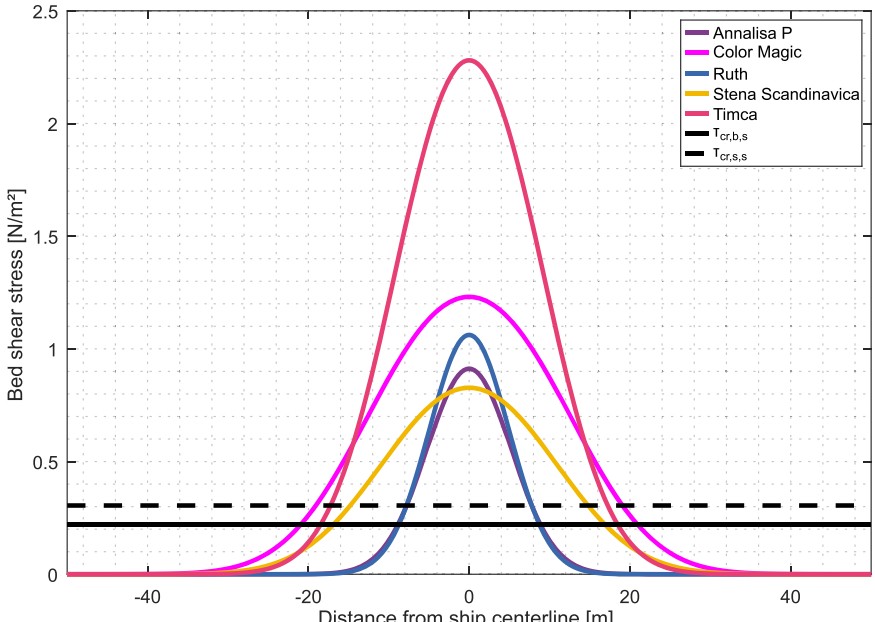

**Fig. 7 | Cross-sectional distribution of propeller-induced bed shear stress at the longitudinal location where maximum stress is sustained.** The curves represent the 5 vessels which we traced with the EK60 fishery split-beam echosounder, with draft, engine power and cruise speed listed in Tab. 1. Horizontal lines mark the critical shear stresses for the initiation of bedload transport ($\tau_{cr,b,s}$) and suspended load transport ($\tau_{cr,s,s}$) of sand. Shear stress values above these thresholds indicate conditions under which seafloor surface sediments can be mobilized in the respective transport regime.

recorded at 2–3 knots perpendicular to the sailing direction of the ships, in the area where the eroded till of unit 3 crops out at the seafloor. The measurements were conducted in areas with 12–16 m water depths. The draught of the passing ships varied between 5.8–9.0 m, and their speed varied between 10.3–15.5 knots (Tab. 1). Crossing between 176 m (Annalisa P) - 416 m (Timca) behind the passing ships, we resolved the wake of all ships in the water column. The sonar volume backscattering strength $Sv$ of the air-bubble-water mixture within the wake reaches values between −70 and −40 dB re 1 m⁻¹. For 3 out of the 5 cases, the echogram shows a propeller wake reaching to the sea surface as well as down to the seafloor (Fig. 6a).

Outside of the wakes from the passing ships, the EK60 data resolves a stratified water column with multiple pycnoclines (Fig. 6a). Adjacent to the ship wakes, the pycnoclines, however, develop a wavy pattern with wave amplitudes locally exceeding 2 m. CTD data collected on the same day as the water column images, but located outside of the wakes of passing ships, also show a multitude of pycnoclines and a strongly stratified water column with about 5 kg/m³ difference between surface and bottom waters (Fig. 6b).

### Propeller jet-induced seabed shear stress
The estimated maximum bed shear stress generated by the investigated vessels ranges from 0.83 N/m² to 2.28 N/m² in a water depth of 13 m (Tab. 1). The lateral cross-section through the respective shear stress footprints shows that bedload transport and suspension of the sandy seabed sediment can be affected in 26 to 61 m wide bands along the track of the ships (Tab. 1, Fig. 7). This width of the zone of active sediment transport agrees with the dimensions of the observed linear ridges which are around 30 m wide and with the width of linear scoured channels described[15] by Krämer et al.

## Discussion
We suggest that meter-scale seafloor erosion of shallow sands and glacial till within unit 3 results from ship propeller-induced wake interaction with the seafloor (Fig. 8). Unit 2, which lies outside of the main shipping corridor, was not subject to propeller-induced erosion and no stones are exposed at its surface. We resolved the wake of all

five ships due to the impedance contrasts induced by the bubbly wake. Between 12–16 m water depth, there was 3–10 m of water column between the seafloor and the ship´s propeller. For 3 out of the 5 cases, the echogram shows a propeller wake overcoming this distance and reaching the seafloor. The most intense water column disturbance was documented for the ship with the closest point of approach (Annalisa P), suggesting that the intensity of the wake decreases over the first couple of minutes after a ship has passed. The intensity and erosion potential of the wake cannot be quantified with the two-dimensional sonar images. The calculated seabed shear stresses induced by the five investigated ships (Fig. 7), however, leave little doubt that the energy released by ship wakes is sufficient to affect both the water column and the seabed.

In the water column, the wake can break through a well-established summer stratification and mix previously separated water masses. At the seabed, the calculated shear stresses exceed the critical thresholds for sediment motion and are capable of initiating bed-load transport and suspension of sand. In the study area, these grain sizes have most likely been eroded from the glacial till by waves[20,30], possibly at lower water levels during the postglacial sea level rise. While now out of reach for mobilization by waves and in the absence of strong currents, they are mobilized by ship-induced shear stress. Where the mobile sand layer is thin enough, this may lead to ongoing abrasion of the glacial till. The wake is thus acting as an erosive agent coming from above, similar to marine mammals and scouring[31], wave forcing in shallow waters[32] or bottom trawling[33]. The location of the depressions within low-permeable till further rules out fluid flow as a genetic origin, which is, for example, responsible for the formation of pockmarks in muddy Holocene sediments in the nearby Eckernförde Bay[34].

The exclusive existence of intense geomorphological changes along the main shipping route supports our interpretation (Fig. 5). The largest vertical changes between 2014 and 2024 occurred within the inward and the outward corridor of shipping, whereas only marginal changes are observed along a band of almost no shipping in-between (Figs. 4–5). The reversed orientation of the steep and the gently dipping flank within the elliptical scour depressions around boulders

**Table 1 | Technical parameters, resulting hydrodynamic loads, and widths of the zones where critical thresholds for bedload and suspended load of sand are exceeded for the five investigated ships**

| Vessel name | Ship type | Draft[a] | Number of pro-pellers | Engine power | Speed over ground[a] (SOG) | SOG (max) | Max. near-bed vel. | Max. bed shear stress | Width sand bedload | Width sand susp. load |
|---|---|---|---|---|---|---|---|---|---|---|
| Parameter | | $d$ | $n_p$ | $P$ | $u_g$ | $u_{max}$ | $u_{b,max}$ | $\tau_{b,max}$ | $w_{s,b}$ | $w_{s,s}$ |
| Unit | | [m] | [-] | [kW] | [kn] | [kn] | [m/s] | [N/m²] | [m] | [m] |
| Annalisa P[b] | Cargo | 8,3 | 1 | 11,200 | 10,3 | 19,0 | 1,56 | 0,91 | 26,28 | 20,98 |
| Color Magic[c] | Ferry | 6,9 | 2 | 31,200 | 13,9 | 22,0 | 2,11 | 1,23 | 54,43 | 45,37 |
| Ruth[d] | Cargo | 9,0 | 1 | 8400 | 15,5 | 18,5 | 1,57 | 1,06 | 27,16 | 21,95 |
| Stena Scandinavica[e] | Ferry | 5,8 | 2 | 25,920 | 14,7 | 22,0 | 1,97 | 0,83 | 41,38 | 33,55 |
| Timca[f] | RoRo | 8,4 | 2 | 25,200 | 15,0 | 22,7 | 2,45 | 2,28 | 60,73 | 51,10 |

Calculations have been conducted for a water depth of 13 m. Information sources.
[a]=from AIS data.
[b]=https://www.hafen-hamburg.de/en/vessels/annalisa-p-47911.
[c]=https://en.wikipedia.org/wiki/MS_Color_Magic.
[d]=https://www.hafen-hamburg.de/en/vessels/ruth-33847.
[e]=https://de.wikipedia.org/wiki/Stena_Scandinavica_(Schiff,_2003).
[f]=https://www.wartsila.com/encyclopedia/term/con-ro-carrier-timca.

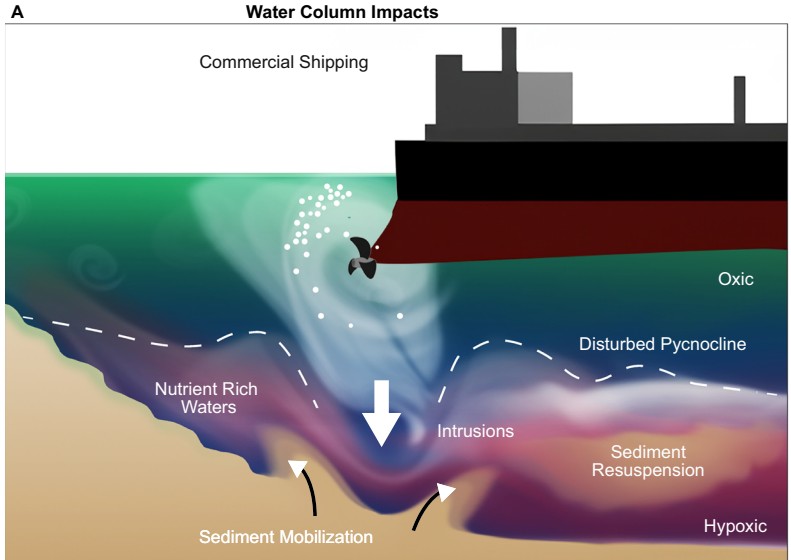

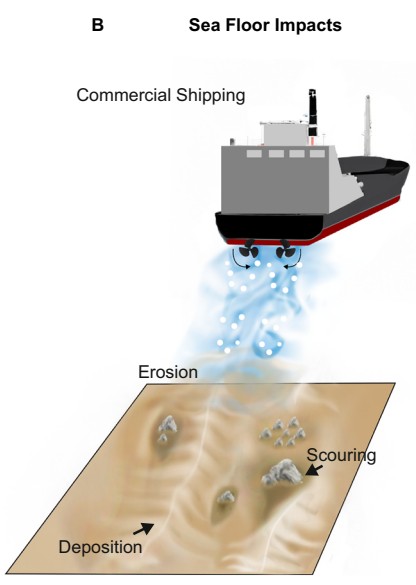

**Fig. 8 | The impact of ship wakes on shallow Baltic waters.** Induced alterations affect **A** the water column and **B** the seafloor and include variations of seafloor morphology, water column integrity, and shallow sediment composition with potential implications for ecosystem functioning and element budgets on a Baltic scale.

(Fig. 3b, c) further indicates a spatially consistent two-directional forcing[35]. The orientation of the flanks matches the sailing direction of the ships. In the outward corridor, with the ships on a northeasterly course where the wake flows in a southwestern direction, the steeply dipping upstream flanks face towards the southwest. In the inward corridor, the side of the gently dipping flanks within the depressions as well as the sailing direction of the ships reverse. Lab experiments have shown that scour holes become wider and longer with increasing rotational speed of ship propellers[13]. The observed morphological directionality, including the sharp boundary in the center, can hardly be explained with currents or ocean circulation that work on more regional scales[36]. Dredging can also be ruled as the genetic origin of the observed morphologies, as no activity is recorded for the area[37,38].

What has not been explained so far is the genetic origin of the northeast–southwest trending linear ridges (Fig. 3a). They are located within the busiest section of the inward corridor of shipping (Fig. 5). Considering only the course of the six ferries that connect Kiel with Oslo, Gothenburg, and Klaipeda shows that the ferries often sail directly above the two ridges (Fig. 5c). Some of the 25 m * 25 m grid

cells have experienced more than 45 ferry crossings over 116 days. We speculate that the ridges develop in the confluence zone of the wake that is generated by two-propeller ships within the two corridors of net erosion underlying the shipping lanes (Fig. 8). All six ferries are moved by two propeller propulsion systems. At water depths around 12–16 m, the wake generated by each propeller will start to interact with the seafloor before interacting with the turbulence created by the second propeller. Penna et al. showed[13] that a counterclockwise rotating propeller (looking in the sailing direction) redirects the sediments to the right. Together, the two propellers rotating in opposite directions may displace the sediment to a central line underneath the hull of the ship, which leads to the formation of the linear ridges. The ridges were most pronounced in the 2024 multibeam bathymetric data. Their more pristine shape in recent years may result from improved auto navigation on the ferries, allowing them to sail exactly the same route on their way towards Kiel. This formation mechanism is conceptually similar to the genesis of sand ribbons[39], which are morphologically comparable to the linear ridges. Sand ribbons are flow-parallel bedforms on sediment-starved shelves and have been observed in the Baltic Sea[40,41].

They form through lateral sediment transport by a secondary circulation pattern transverse to the dominant current direction, induced by variations in seafloor roughness. Mobile sand accumulates in areas of lower bed shear stress[42].

Till erosion represents one of the main sources of sedimentary particles in the Baltic Sea. While erosion along coastal cliffs has been studied and partly quantified by different authors[22,23,43], little is known about the abrasion rates and processes that govern erosion of till at the seafloor[24]. With more than 16 m, the Bay of Fundy, Canada, hosts the highest tidal range on the planet. Here, Wu et al. and Shaw et al. showed[44,45], that glacial till is eroded in areas where near seabed flow velocities exceed 3.5 m/s. With flume experiments, Mier and Garcia further showed[46] that glacial till is eroded above a shear stress of 4.2 N/m². The calculated flow velocities (Tab. 1) and shear stresses for the selected ships are lower than these thresholds. Nevertheless, the ship-induced shear stresses are obviously high enough to cause bedload transport of sand (Fig. 7) and initiate abrasion of glacial till. A similar process is already known from abrasion platforms in shallow water environments in the southwestern Baltic Sea, where it occurs due to natural forcings[32,47]. For the area of our study, model results show that the naturally occurring combined wave-induced and current-induced shear stresses are, however, far too low to initiate motion of sands[48,49] and remain far below the levels induced by ship propellers.

The fate of the eroded sediments differs depending on their grain size. The large boulders remain in place and interact with the currents from the ship's wake. Therefore, they serve as nuclei for scouring and provide a positive feedback mechanism for seafloor erosion[50]. The finer-grained clay material on the other end of the grain size spectrum, which is also eroded from the till, is suspended and transported away by natural currents. The sandy fraction is redeposited within the area, partly forming sorted bedforms[15]. This is exemplified by the shipping parallel (northeast–southwest) elongation of the fields of sand dunes and the perpendicular strike-direction of their crests, which does not match the local current or fetch regime[51]. Redeposited sands also build the linear ridges as well as the smooth patches in between the depressions, from which the fields of sand dunes develop[51] (Fig. 4). The observed maximum depth changes in the sand, of up to 1.5 m over 10 years, outpace natural sedimentation rates in the Baltic Sea[52].

To estimate the total area affected by shipping-induced sediment alteration on a Baltic scale, we used the HELCOM AIS shipping density map from 2022[2]. We only considered grid cells that were at least crossed by one ship per day on average (i.e., minimum of 365 crossings for the entire year). We further only considered regions shallower than 20 m, as the likelihood of the wake reaching the seabed obviously decreases with increasing water depths. Together, this covers an area of 30,356 km², which is about 7.5% of the entire Baltic Sea. The upscaled magnitude of alteration on a Baltic scale ranges at a similar level compared to the 6.91% reported[18] by Díaz-Mendoza et al., who documented propeller-induced bedforms and scour features, primarily on sandy substrates. The volume of the depressions within unit 3 sums up to 0.063 m³/m². Using this value as a benchmark would account for a total eroded volume of 1.9 km³ for the entire Baltic Sea. We note that a single ship crossing will not lead to meter-scale seabed erosion. However, the Kiel Canal, for example, was finished in 1895, and since then hundreds of thousands of ships have passed through the study area as well as to other parts of the Baltic Sea. Their cumulative erosive potential on decadal to centennial timescales is high enough to remove and erode meters of sand and till. It is obvious that till is not the exclusive strata in the Baltic Sea in regions where the water depth is shallower than 20 m. Ship wake-induced abrasion and redistribution of muddy sediments, or in places where bedrock crops out at the seafloor, have not been investigated so far and will require similar field campaigns within shipping corridors elsewhere in the Baltic Sea.

In many regions of the Baltic Sea, commercial shipping is intense, whereas the water depth is relatively shallow (Fig. 1). Here, the ship wake has potentially substantial, yet largely unrecognized, implications for biogeochemical cycling. In the water column, we show that propeller wakes actively disrupt even a strong vertical summer stratification and mix previously separated water masses. Furthermore, the vertical displacement of stratified water masses can locally excite internal waves, visible as a wavy pattern in our sonar data (Fig. 6a). This may further increase the possibility of diapycnal mixing by generating increased vertical shear[53] or wave breaking[54]. Together, this provides direct evidence of the mechanism linking ship-induced water column disturbance to the recently reported increases in methane emissions[11]. Further, turbulent mixing likely temporarily decreases nutrient limitation in surface waters, potentially leading to localized enhancements in phytoplankton growth[55]. Towards greater depths, the physical disturbance may enhance the intrusion of oxygen into seasonally or permanently anoxic bottom waters, altering redox-sensitive biogeochemical processes such as denitrification, sulfide release, and trace metal cycling[56]. Episodic oxygenation of suboxic or anoxic bottom waters can further disrupt vertical redox gradients, creating transient biogeochemical niches and influencing microbial activity, phytoplankton composition, and zooplankton behavior[3].

In the sediments, the erosion of till and redistribution of overlying sands mobilize fine particles and associated organic matter, enhancing benthic-pelagic coupling and modifying carbon and nutrient fluxes akin to bottom trawling[57,58]. Exchange processes in sandy sediments are driven by pressure gradients induced by overlying currents and waves[59–63]. Ship passages may induce transient pressure gradients that enhance sediment-water exchange, while at the same time sand mobilization could reduce exchange[64]. However, which of the processes dominates remains unexplored. Additionally, in benthic habitats, the physical instability of the sediment surface can disrupt the establishment of structured infaunal and epifaunal communities, favoring opportunistic and disturbance-tolerant species while displacing more sensitive taxa[65]. These shifts can cascade through sedimentary biogeochemical processes, altering bioirrigation, bioturbation, and benthic-pelagic nutrient fluxes. These disturbances may further impact higher trophic levels through changes in food availability, water clarity, and chemical habitat conditions.

Our study suggests that ship wakes have the potential to erode sedimentary strata, mix oxygen, nutrients and other elements within previously stratified waters, alter turbidity and light availability at depth, and mobilize large amounts of sand. Together, these processes can reshape marine ecosystems and element budgets on a Baltic scale. Sandy sediments, the dominant substrate in most coastal settings, are particularly vulnerable, as their exchange processes are tightly coupled to pressure gradients and flow dynamics. Yet, these habitats and their role in mediating benthic–pelagic exchange remain underrepresented in current research. Despite the potential significance of ship-induced impacts, the cumulative ecological consequences of this persistent anthropogenic forcing remain largely unresolved and warrant targeted, interdisciplinary research to quantify long-term impacts on ecosystem function and biogeochemical cycling at basin scales. Wake-induced vertical changes may require adjusted and customized monitoring strategies and the update of the underlying seabed geodata. It should be investigated whether minor adjustments to navigation routes could offer a practical means to counteract the genesis of sand ridges along their tracks.

## Method

### Multibeam bathymetric data

Multibeam bathymetric data analysed in this manuscript were collected in 2014 by the German Federal Maritime and Hydrographic Agency (BSH), and 2024 by the Marine Geophysics and Hydroacoustic Group of Kiel University during RV ALKOR cruise AL619[25]. The BSH data were gridded to a resolution of 1 m.

In 2024, a NORBIT Multibeam iWBMSc system was used with a frequency of 400 kHz, a theoretical range resolution of 9 mm and a beam width of 0.9 * 1.9°. Deployed from the moonpool of RV Alkor at a draft of 4.5 m, this results in a spatial footprint of 14 * 29 cm in a water depth of 13 m. The roll offset of the system was calibrated by collecting data along the same profile in opposite directions. To account for the spatially and temporally inhomogeneous physical structure of the water column within the area of frequent mixing, the sound velocity of the water column was measured repeatedly during multiple Sound Velocity Profiler stations and used for correcting the soundings for refraction. The corrected soundings were spline filtered and manually edited in QPS Qimera 2.6 to account for the complex seafloor morphology within unit 3. Motion and navigation data were left unprocessed except for occasional Real Time Kinematic (RTK) height outlier removal. The processed soundings were exported and interpolated to a 0.5 * 0.5 m raster using MB-systems using a spline with tensions algorithm[66]. The final grid is projected in UTM32N and tidally reduced to mean sea level using RTK height and the German Combined Quasigeoid (GCG2016).

For the morphological analysis of the bathymetric data, we used the software ArcGIS. The data sets were resampled to 1 m and then subtracted using the raster calculator to create the differential bathymetry maps. Semi-automated mapping of the depressions was conducted through a combination of multiple hydrogeology tools in ArcMap. It mainly followed workflows used for mapping of seabed pockmarks[29]. All depressions were filled up to their respective pour points. The newly created grid was then subtracted from the original grid, giving the depths of the depressions. Areas that had changed by more than 1 cm were classified and outlined with polygons. The total volume of the depressions as well as the mean change in the study area (red outline, Fig. 2b) were calculated using the GIS Zonal statistics tool.

### EK60 water column data
During RV ALKOR cruise AL619 (2024), the ship's SIMRAD EK60 system was used to map the wake of passing ships in the water column[25]. The split beam echosounder, originally designed for fishery acoustics, is optimized for water column analyses and capable of tracing the spatial extent and duration of wake-induced water column disturbance. It was operated with three transducers sending out a constant wave pulse centered around 70 kHz, 120 kHz, and 200 kHz. Best results were achieved using 120 kHz with a 6.7° opening angle, pulse length was set to 128 μs to achieve a high range resolution, the ping interval was set constant of 20 ms. Other, potentially interfering sonars were switched off during the water column recording.

### Automatic identification system (AIS) data
We used AIS data of the Baltic Sea provided by the Danish Maritime Authority (https://www.dma.dk, last accessed 17 December 2024). For our analysis, we considered 116 days (1.6.–24.9.2024) of AIS data prior to the collection of the 2024 bathymetric data. We first linked all AIS data points of each vessel to a continuous track, before counting the number of tracks in 25 * 25 m grid cells. This allows us to derive the absolute number of crossings for each grid cell (Fig. 5). We excluded ships with less than 4.5 m of draught and those sailing at less than 2 kn from our analysis. The draught filter was meant to exclude sailing vessels and other non-commercial vessels in the study area. While we note that there are lots of commercial vessels with a draught of less than 4.5 m, the overall picture, however, gets incredibly busy if small commercial and sailing vessels are included, which tend not to sail along the main shipping corridors. The speed cutoff was intended to avoid anchoring ships. According to international regulations, all fishing vessels exceeding 15 m in length, all passenger vessels, and ships with a gross tonnage greater than 300 tons are required to carry class A AIS transmitters. Smaller vessels, such as recreational boats, may use class B AIS transmitters, but are not obliged to do so.

To upscale the erosion rates documented in the Bay of Kiel to the entire Baltic Sea, we used the HELCOM AIS shipping density map from 2022[2]. This dataset represents the density of all IMO registered ships operating in the Baltic Sea. Shipping density is defined as the number of ships crossing a 1 ×1 km grid cell.

### Sedimentological data
On board the cruise AL619, we took six VanVeen-Grab samples from which we used the upper 2 cm for grain size distribution analysis. The finer samples were measured with the Malvern Mastersizer 3000 Aero S, which scans a spectrum of 0.01 μm to 3000 μm. The unsorted samples AL619_55 and AL619_58-2 were analyzed with the Retsch AS 200 control sieve system between 63 μm and 20 mm.

### CTD data
To determine changes in water density with depth, an AML-MINOS-X probe was used as a CTD sensor during cruise AL619, from which density was calculated[67].

### Critical shear stress of surface sediments
Critical shear stresses for sediment motion and suspension were derived from grain size distributions obtained from grab samples. The first mode of the grain size distribution was used to characterize the sediment and compute the dimensionless grain diameter[68]. Using this parameter, threshold Shields numbers for initiation of bedload motion and suspension were calculated[69] following van Rijn and converted into critical shear stresses for bedload and suspended load transport. Salinity and temperature data from a nearby long-term mooring were used to constrain the kinematic viscosity required for this conversion. For a full description[15] of this parameterization, see Krämer et al.

### Propeller-induced bed shear stress
Near-bed velocities and associated bed shear stresses generated by ship propeller jets were estimated from empirical relations describing the jet efflux velocity and its axial and radial decay[70–72]. Propeller diameters were derived from ship drafts when not available, and efflux velocities were estimated from vessel engine power and operational speed[73,74]. For moving vessels, reductions relative to the seabed flow were accounted for, and velocity fields were translated into bed shear stress footprints using a quadratic stress law with an empirical friction factor. For twin-propeller vessels, contributions from individual drives were computed separately and superimposed. Ship dimensions, speeds, and engine power were obtained from technical specifications and AIS data. A full description[15] of equations and parameterization is provided by Krämer et al.

### Reporting summary
Further information on research design is available in the Nature Portfolio Reporting Summary linked to this article.

## Data availability
The bathymetry data of 2014 were collected by BSH and are freely available via the SeaDataNet Pan-European infrastructure for ocean and marine data management (https://www.seadatanet.org). Automatic identification system (AIS) data are archived by the Danish Maritime Authority (https://www.dma.dk), which provides free access to historical AIS raw data and information on AIS usage. The 2024 bathymetry data[75] and the EK60 data[76] are available from https://doi.iow.de/.

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

## Acknowledgements
Funding was provided as part of a small strategic institute expansion at the IOW (Shore to Basin, S2B). We thank the captain and crew of R/V Alkor for their professional support during cruise AL619. The onboard pre-processing of the multibeam and split-beam data was kindly supported by QPS, providing student software licenses. The study contains AIS data from the Danish Maritime Authority that is used in accordance with the conditions for the use of Danish public data. This research has been supported by the Leibniz Research Network "Earth & Societies".

## Author contributions
J.G., L.R., L.B., S.K., and J.S.v.D. collected and processed the 2024 bathymetric data. P.W. provided the processed 2014 bathymetric data. J.S.v.D. collected and processed the EK60 water column data. K.K. conducted the propeller-induced seabed shear stress calculations. P.F. processed the AIS data. L.B. collected the grab samples and analyzed the grain size distribution. L.R. conducted the quantitative morphologic analyses. S.A. analyzed the biogeochemical implications. F.T. analyzed the detailed seabed morphologies within the shipping corridor. J.G. lead the writing of the manuscript with all authors contributing.

## Funding

## Competing interests
The authors declare no competing interests.
