## [Transparent Peer Review file · Nature Communications]

Ship wake induced water column mixing and meter-scale seabed erosion in the Baltic Sea

Corresponding Author: Dr Jacob Geersen

Version 0:

Reviewer comments:

Reviewer #1

(Remarks to the Author)

Geersen et al. describe the results of two surveys of the seabed in the Baltic Sea. The data reveal interesting bedforms, including scoured depressions and sand dune trains. Comparison with shipping position data strongly suggests that shipping induced water movements are the cause. One water-column sonar profile supports this interpretation.

I enjoyed reading the article and think it is publishable after some revision. However, I wondered if the authors have picked the right audience for this. The data do not reveal the biological and geochemical effects that they claim (rather, they are inferred). However, I wondered if there could be an important statement made about how glacial tills erode. To explain this, please look at Shaw et al. (2012). The Canadians collected some very nice data around the Bay of Fundy, where the extreme tides lead to extreme currents. Their data revealed scours where tidal currents are elevated around passages, reaching 10s of m in relative depth. Notably, the margins of the scours are sharp, not blurred.

Erosion of a sediment might be expected to vary a lot spatially if the sediment has a wide range of grain sizes or if the maximum current at any location has varied a lot over time. Hence, sandy areas tend not to have such abrupt scour margins. If the till is clay-rich, presumably there is a shear-stress threshold above which particles can be extracted by the current. If impacts of saltating particles are important for loosening the other seabed particles, there might also be threshold at which saltation starts, depending on the grain size. Whatever the cause, the scours strongly suggest a threshold shear stress is present. From Shaw et al.'s Figure 15, it should be possible to evaluate the stresses involved (it would involve seabed roughness, which is poorly known, though could be estimated from the seabed photos). Their scours are much deeper and we'd expect till to become stiffer with compaction (Skempton). But I wondered how their stresses might be compared with those in your area. From the internal wave suggested in Figure 5, it might be possible to work out water velocity. Alternatively, a jet from the propeller might generate higher velocity at the seabed, though I'd be surprised if it were as large as those tidal currents from the Bay of Fundy. Hence, the Bay of Fundy results provide a maximum. There should be geotechnical measurements of in situ stress also to compare against.

For the reader's understanding before continuing to the data, I'd suggest some background text on the tidal and wind-blown currents and the waves. Waves should mostly not be important, though extreme waves from an easterly wind may conceivably be large enough to affect the bed in 10-20 m of water. Is the combined stress from these with the propeller effects important?

Lines 20-24. In my opinion, including sentences like these on potential future work leaves a poor impression (it comes across as you seeking funding), while the reader is then distracted from your results. If you remove these sentences, the abstract will be much sharper and leave room for other aspects of your study. Think how the last impression of your abstract helps to motivate a reader to cite its findings.

At risk of promoting myself (hence this is not a request to cite), the authors might look at Mitchell et al. (2021) - your results suggest how the stratigraphy from human activity will develop, hence perhaps a link between your results and the Anthropocene debates. In Mitchell (2014) and Mitchell et al. (2012), we looked into whether or how flows (turbidity currents and tidal currents) lead to erosion. Some of it may be useful for the calculations. I was still not sure if the underlying till (pre-erosion) is expected to be clay rich or not and how cohesive. Could some more background on it be included?

Detailed suggestions:

Line#

18-19 ...stressor are unquantified, though will include modifications to planktonic life, benthic habitats,...

28 Unclear what type of catchment area. I suggest not using this as it can be confused with rainfall catchment.

32 the regions ... -> 56% of the region is shallower than 20 m. (if correct? "below 20 m" is ambiguous)

39 Marie Tharp's article is very broad, though isn't there a publication on Kiel's historical hydrographic surveying? That's more relevant here.

45 "in action"? It is not possible to measure during erosion. I'd leave this out.

forces -> influences

60. This case study area has bathymetry shallower than 20 m and is an area of intense commercial shipping activity due to the entrance to the Kiel Canal, one of (reference if there is one?).

67 delete "which represents the landform"

71 under -> by

78. Readers in my experience seem to like a traditional structure, so you might put this first paragraph in a section marked "Methods and data".

81 allow us to

It would be useful to know a bit about how the data were collected and processed. For example, were corrections made for survey vessel draft (varies with use of marine fuel) and tidal heights? Can you estimate how accurate your values were, as this affects how well changes can be detected?

You might mention the survey direction and the artifacts, which are presumably due to small biases in roll corrections mainly. Hughes Clark et al. might be cited for along-track artifacts. Although the method might not be applicable if your survey lines are not coincident, see Schmitt et al. - sorry for the self-promotion, though it may be useful for some ideas.

Is there a web link for the AIS data?

88 Please specify the system used and its acoustic frequency.

For some general thoughts on how the magnitude of acoustic backscattering might vary and what might cause backscattering (if fish or particles or bubbles caused by propeller cavitation, or other water fluctuations in density/velocity), see Clay and Medwin and Stanton and Clay. There will likely be more recent references.

90 and was subdivided into

91 locates -> is

92 ...scales, hence it has low roughness (F...).

94 that is elevated

95 The boundary between unit 1 and the other units is gradual.

99 Unit 2 occurs along

100 between 100 and 12 m water depths.

101 On a regional scale -> Regionally

103 intersected -> crossed

southwest-striking band

Please mention the roughness value (make your text more quantitative).

104-106 The rough ... - this is a bit vague, I'd suggest deleting this sentence as it is not needed.

110 trains of sand dunes with crests striking NW-SE.

112 that elevate -> elevated

114 ... and lie oblique to survey directions, hence are not a result of survey artifacts.

116 It might be better to refer to only Figure 3b and 3c, and mark the smooth patches. Do you mean the areas marked "smooth seafloor"? It is not obvious what you mean by the sand dunes developing from them - I suggest leaving out this comment (how do you know?).

117 Rather than asserting that unit 3 comprises sand, talk through the grain size measurements first, then say why you think the area more broadly comprises sand (what evidence? - is this marked on hydrographic charts?).

119. Here you are putting interpretation before observation. Talk through the distribution first then what you infer from it.

122 compare? Do you mean based on or similar to?

123. I don't know from this text what you have done to get these volumes exactly, consequently it is difficult to interpret the values. I suggest including a short section on the method used within your Methods section.

124 "sums up to" -> is

125, which represents 0.355... over the whole area of Unit 3

(if correct- it is unclear what area your 7.2 km² represents, is it the whole of Unit 3 or only the depressions?).

126 shape -> in plan-view

126 striking NE-SW.

127 have asymmetric profiles, ...

129 remains stable -> is uniform

130 The orientations ... varies across the study area.

(if correct - unclear)

131 ...the steepest flank..

132 This tendency reverses. There, the steep...

138 Delete "wide"

139 concentrates on -> overlies

141 The 2024 data reveal NE-SW-trending ridges that were not so prominent in the ...

143. 0.13 m value is missing uncertainties.

148-152. Delete this paragraph here, as you need the reader to have studied the AIS data first. Instead this should go in your discussion section.

(149 concentrated)
154 concentrates -> is concentrated
156 hardly -> rarely crossed by ships.
157 Between 1.6.2024 and 24.9.2024, 1313 ships returned their locations through the AIS in the study area.
(please make the text easier to follow)
159 delete "therefore"
161 ... 5648, representing 49 passages per day.
163 The maximum draft (10 m) corresponds nearly with the maximum... (9.5 m). Only four vessels with drafts >10 m sailed through the area, likely corresponding to berthings at the Port of Kiel, which currently ...
167 Water column acoustic backscattering following ship passage
168-170. the capabilities of the sonar should be in methods. Use this section purely for observations.
172 were conducted in areas with 12-16 m water depths.
174 please delete "we clearly" Talk through the observations first, then say how you interpret them.
175 Figure 5, for example, shows....
180 locates -> lies
182 Sentence can be deleted.
184. Reader cannot evaluate this. Is it possible to include the other profiles as an electronic supplement?
185 "The intensity ..." - I'm not sure this is true. If you know the wavelength and amplitude of the waveform, I think you can estimate a current speed. For surface waves for comparison this is possible (Masselink and Hughes).
189 I'm not so sure it would be similar to bottom trawling. A better analogy might be simply the effects of surface waves in shallow water.
210 by the two corridors of erosion underlying the shipping lanes (Fig. 6).
224 sedimentation -> sedimentary particles
226 There has been quite a lot of work done by the group at the UK's IOS, including Neil Kenyon, Arthur Stride, Bob Belderson and others. It is fair to say it was largely descriptive (non-quantitative).
234. The Berne et al. citation is misleading. I think you need to have outlined the tidal currents, wind-blown currents and surface waves as background or part of introduction.
236 The depth changes in the sands here, of up to, outpace ...
338 This statistic has been mentioned before, so rephrase this.
246 ... 1895 and since then hundreds as well as to other parts of the Baltic Sea.
248 Delete sentence (Many of them..) unless you have some values on these drafts. E.g., were there ships, such as battleships, of sufficient size to merit mentioning?
257 Towards greater depths, ...

281. I guess the format of the journal leads to the methods being here. In my opinion, this leads to confusion as the reader has to flit back and forth to understand the study in detail. I would have submitted this work to a journal with a more traditional format.
285 delete "provided in xyz format and" using the GMT software modules
291 used for correcting the soundings for refraction.
292 spline is an interpolation - not sure what you mean.
296 You need to explain what UTM means.
How were the tidal height data recorded (tide gauge, if so where, or model?).
303 depths of the depressions.
You need to explain also how the roughness in Figure 1B was calculated (roughness depends on the spatial scale of measurement and the mean surface that it is measured against).
319 Were there any vessels with drafts <4.5 m travelling at exceptional speed?
332-3 This kind of information is needed to interpret the figures shown, so it is strange having it here. Hence, in my opinion, a more traditional format is better.
335. It would have been nice to know a bit more about these samples. E.g., was the rest of the sediment similar in terms of grain size? What were the maximum particle sizes present? Were any clays present?

439-440. I suggest putting the things in brackets at the end of the sentence to make it more easily read. Hence: "The traffic Canal (green and brown arrows, respectively)."
442. You are mixing up observations and interpretation here. I would simply highlight the high roughness coinciding with sea lines.
442 "Flat and featureless unit 1..." The reader cannot see unit 1. Again, I would simply highlight that this area has low relief or leave this out (obvious).
447. Somewhere you might mention the survey orientation and that the NW-SE lines are artifacts of the data (for people without our backgrounds).
456 Differential -> Bathymetric
459. It might be good to clarify what the phi scale means (for non-geologists). Values in mm might also be useful (second horizontal axis).
460 Selection of data showing how two areas changed bathymetry between 2014 and 2024.
463. Please mention in the caption what these intensity values mean (i.e., simply give min/max with units for B and C).
Figure 4C - difficult to work out the water depth from the colour table shown. I would guess water depth is around 15 m?
Figure 5 - The undulating reflections/backscattering layers are not pycnocline exactly. Rather they are due to fish and other causes of acoustic impedance variations within the water, which are being moved up and down by an internal wave acting on the pycnocline.

Figure 6 in B, you mention disturbed thermocline, though here it should be pycnocline.
474 Figure caption does not make sense. B does not show sediment biogeochemistry.

I enjoyed reading the article and look forward to reading it in print.
Neil Mitchell, University of Manchester, May 2025.

- Clay, C.S. and Medwin, H., 1977. *Acoustical oceanography: principles and applications*. Wiley Interscience, New York, 544 pp.
- Hughes Clarke, J.E., 2003. Dynamic motion residuals in swath sonar data: Ironing out the creases. *Int. Hydrog. Rev.*, 4: 6-23.
- Masselink, G. and Hughes, M.G., 2003. *Introduction to coastal processes and geomorphology*. Arnold, London, 354 pp.
- Mitchell, N.C., Huthnance, J.M., Schmitt, T. and Todd, B., 2012. Threshold of erosion of submarine bedrock landscapes by tidal currents. *Earth surface processes and landforms*, 38: 627-639.
- Mitchell, N.C., 2014. Bedrock erosion by sedimentary flows in submarine canyons. *Geosphere*, 10: doi:10.1130/GES01008.01001.
- Mitchell, N.C., Jerrett, R. and Langman, R., 2022. Dynamics and stratigraphy of a tidal sand ridge in the Bristol Channel (Nash Sands banner bank) from repeated high-resolution multibeam echo-sounder surveys. *Sedimentology*, 69: 1051-1082.
- Schmitt, T., Mitchell, N.C. and Ramsay, A.T.S., 2008. Characterizing uncertainties for quantifying bathymetry change between time-separated multibeam echo-sounder surveys. *Cont. Shelf Res.*, 28: 1166-1176.
- Shaw, J., Todd, B.J., Li, M.Z. and Wu, Y., 2012. Anatomy of the tidal scour system at Minas Passage, Bay of Fundy, Canada. *Mar. Geol.*, 323-325: 123-134.
- Skempton, A.W., 1970. The consolidation of clays by gravitational compaction. *Q. J. Geol. Soc. London*, 125: 373-411.
- Stanton, T.K. and Clay, C.S., 1986. Sonar echo statistics as a remote-sensing tool: volume and seafloor. *IEEE J. Oceanic Eng.*, OE-11(1): 79-96.

Reviewer #2

(Remarks to the Author)

This research article presents a really interesting and new perspective on ship wake erosion. The combination of repeat multibeam surveys, geologic sampling, water column imagery and morphological analyses provides compelling evidence for the erosion and redistribution of sediment and hard substrate, in areas with heavy vessel traffic. There are some general and specific comments I have made below, which I hope are helpful to the authors in revising this manuscript. I feel there are some things that would benefit from revision, but they are mostly related to manuscript structure rather than methods. There are some sentences that were a bit confusing as a reader, which I have highlighted for revision in the specific comments section.

General comments:

- The introduction is a bit confusing to me. It provides quite a lot of detail about the Baltic, which is fine. But I feel it could be improved to start with an introduction about shipping impacts, and then move into the Baltic as a case study. I think the introduction needs to provide more about the state of the literature on shipping wake impacts. As a read There is a lot of detail about the Baltic formation and geology, but it lacks information about the shipping context up front, for example where is the Kiel Canal?
- The figures are generally well compiled and clear. But there are a few suggestions in the specific comments that would help those not as familiar with the Baltic geography to understand spatial interactions, or moving figures forward or back.
- I think the change detection work is great. But please make sure you aren't providing more accuracy than you are able to with these systems? You mention you can see less than 20cm (line 136) but then report things to centimetre difference.

Specific comments:

Line 31 – "In the southwestern Baltic Sea (west of 15°E), the regions below 20 m even account for 56% of the entire area."

This is a confusing sentence? Perhaps revise?

Line 36 – "The first bathymetric measurements with an echosounder, interestingly already conducted in the Bay of Kiel, took place over 100 years ago." Is this relevant to this paper? Perhaps more focus could be placed on shipping impacts over time rather than mapping the Baltic?

Line 48 – this reads more like an introductory sentence to me, the paper is about shipping impacts, with a demonstrated case study in the Baltic. Perhaps this is the way to structure the introduction?

Line 50 – one thing that has been quite extensively studied for a long time is ship noise, perhaps worth mentioning that in this review section? I realise this isnt about noise, but surely this would also be an impact here.

Figure 1 – it feels like this figure is missing information about the shipping intensity. I see it is shown in Fig 4A, is it possible to bring this into Fig 1 to clearly show why the Baltic is a good place to study ship traffic impacts?

Also, can you add a colour scale onto the figure so we know what the inset depth range is? Or is it the same as 1A?

Fig 1B – I like that you show the roughness, but its not clear why this is needed in Fig 1? It seems to be more of a morphological analysis and could be placed later. That way you could bring shipping intensity into this figure and make the context inset bigger and with more labels.

Paragraph 59-76 – the introduction about the Baltic generally, from paragraph 1 could fit nicely into this paragraph.

Line 61 – can you label the Kiel canal on Fig 1? Those unfamiliar with this region may not know where this is.

Line 71 – is the Bay of Kiel and Kiel Bay the same place?

Line 76 – any sort of survey in particular?

Line 76 – extra “,” that isnt necessary

Line 78 – This sentence is missing something? “All hydroacoustic and geologic data locate in the central Bay of Kiel”

Line 82/282 – were the 2014 and 2024 data collected with the same or similar systems? And what frequencies were these systems using? This could be important when doing change detection as if you have a vastly different frequency then the changes could be due to that (somewhat) as well as erosive mechanisms

Line 87 – which type of echosounder was used for this?

Figure 2 – the depth colour scales appears to change from 2A to the insets, it might be clearer if colour scales are kept the same?

Fig 2(B) - the label says 2012 bathymetry, which is this? You have not described use of a 2012 dataset? Is this a typo?

Fig 2C – this graph seems to show that both the blue and red depressions have slope aspects the vary from SW to NE. I see there is a peak where higher slope blue depressions are more to the NE and red depressions with a higher slope are more SW facing. But perhaps a statistical summary of this would be helpful to show that this difference can be quantified? And/or a density plot showing the distribution of points? To help with more clearly articulating this point. Its quite hard to see what you describe and so its not fully convincing to me just yet.

Line 89 – I think unit should be singular, or you could say description of morphological units.

Line 108 – is there any seafloor backscatter to show the substrate variation and if these “stones” are relatively hard or different substrate compared to surrounds?

Line 124- “Their cumulative volume sums up to 2,556,000 m³, which accounts to 0,355 m³/m².” this sentence is unclear to me. I don’t understand how we got to a volume over an area, please explain how you determined the volume of the pockmarks (was is individually determined or their area multiplied by the average depth?) and help me understand why this is relevant?

Line 134- you mention the limits of detection here but in the scale bar in Fig 3a, you have the differencing down to 2 decimal places. Perhaps its worth only having one decimal place in the figure given the detection limit? Similarly you report mean depth changes of 0.13 m in line 143, should this be just 1 decimal place?

Line 137 – perhaps this title should be “repeat” instead of time-lapse? To me time-lapse signals many instances, whereas repeat is resurveying the area at a different time-step. Its finicky I know but perhaps more accurate.

Figure 3 – to my eyes the 2014 and 2024 datasets appear to have different resolutions, with the 2024 looking to be higher resolution than 2014 – which makes sense given the respective ages of each. Perhaps in this figure caption or by labelling, its worth mentioning these figures show different resolution data but that you did the differencing with the same resolution?

Line 154 – perhaps “lane” is a better word than “stripe”?

Line 157 – can you also provide the duration on this timeframe?

Line 157 – can you rephrase this to say “ 1313 individual ships were identified using AIS, with a draft of...”

Line 158 – can you quantify the max number of times a single ship passed in 1 day

Line 160 – does the AIS data have information about vessel tonnage?

Line 162 – what does sub-weekly mean?

Lies 160-166 – this is a bit confusing, maybe I misunderstood – but you say that the max draft is 9.5 then say 4 ships came in with 10 m draft because the max draft is 11.5m. I think maybe because im not familiar with where the canal is relative to the port and so could this be more clearly explained and perhaps add these locations onto figures so people can follow the geography?

Line 168 – change “towards the seafloor” to “in the water column”

Line 169 – this sentence has already been mentioned above I think “It is optimized for water column analyses and capable of tracing the spatial extent and duration of wake induced water column disturbance” – and its probably more suited to methods too?

Line 171 – how far (in meters or distance measurement) behind the ships were you approximately?

Line 180 – change to “...consists of till but is located outside the main shipping corridor..”

Line 181 – the word “major” seems a bit ambiguous as to what you mean. Does that mean there was some erosion but not much? What is it from? I suggest changing “major” to “propeller-induced” or similar

Line 185 – just out of interest, any idea how long after the ship has gone the water remains disturbed?

Line 202-204 – I agree but I feel like if you mention this it would be good to mention what the oceanic conditions are in the region?

Line 205 – minor grammar, should read “they ARE locateD within the busiest section”

Fig 6 – I really like that the authors have brought everything together but im not sure what panel A adds to this figure? I think B and C provide the process you are trying to describe well. Perhaps you could just reference your earlier figure 4 or 2 in the caption instead of repeating it in this one?

Line 210 – do you know how many of the ships would have two propellers?

Line 214 – it would be great if this was included in Fig 6, how the ship directionality impacts the turbulence and sediment redistribution. “Together, the two propellers therefore displace the sediment to a central line underneath the hull of the ship, which leads to the formation of the linear ridges.”

Line 232 – are these “sorted bedforms” the ridges?

Line 243 – I agree but wouldn’t deeper waters also have bigger ships?

Line 248/9 – perhaps change this to “to cause turbulent flow down to at least 20 m water depth” to simplify?

Line 270-272 – its not clear if this is from your study or from the literature generally? Please specify and add references as needed

Line 277-279 – is there more we should investigate in these substrates too? Perhaps adding a suggestion to look at infaunal community disturbance or noise pollution or similar would be a nice end suggestion for future research direction?

Line 291 – were you able to do an SVP in the wake at any point?

Line 304 – don’t you say somewhere else that you can only see changes >20 cm?

Line 475 – does this figure show sediment biogeochemistry, or ecosystem function? I realise these are impacts of the processes you describe in the paper, but this figure isnt really showing that, its showing the physical process of wake

erosion only (in water and sediments). Perhaps refine the caption to reflect what the figure shows?

Reviewer #3

(Remarks to the Author)

Summary

The manuscript "Ship wake induced water column mixing and meter-scale seabed erosion in the Baltic Sea" by Geersen et al (Ref.NCOMMS-25-33870) provides new insights into the impact of commercial shipping on the morphology of the seafloor in shallow-water environments, particularly in areas with heavy marine traffic. We find the topic and the idea of the manuscript of scientific interest. However, we believe that additional evidence and further development are needed to reinforce several key aspects of the analysis and interpretation, and to strengthen the manuscript's conclusions. We therefore encourage the authors to consider the following recommendations and revise the manuscript accordingly.

This review report discusses two key issues relating to the interpretation of erosive seafloor features. It offers suggestions, asks questions and highlights textual corrections to improve the clarity, coherence and robustness of the manuscript.

Major weakness

- The authors provide a detailed characterisation of the erosional bedforms generated by repeated ship propeller passage in Kiel Bay. These features are of great interest as they provide valuable insights into the impact of human activity on seafloor morphology and sedimentary cover. However, the manuscript would benefit from clearer explanations of its novelty and of how it builds upon previous work in the region.

Specifically, similar bedforms have recently been described in the south-western Baltic Sea (Díaz-Mendoza et al., 2023; 2025). Díaz-Mendoza et al. suggest that propeller-induced features, such as small subaqueous dunes, scour pits and comet marks, were found to affect 6.91% of the impacted seabed, particularly over sandy substrates with boulders (see Figure 5 and Table 3 in Díaz-Mendoza et al., 2025). However, the introduction by Geersen et al. only briefly mentions these studies, without clearly explaining how the present work builds upon, complements, or differs from them.

Therefore, a more thorough discussion in the introduction of existing evidence of similar morphologies in this or other regions, and of the specific new insights provided by the present study would help to situate this work more clearly within the broader scientific context, clarifying its contribution to advancing knowledge in this field.

One of the study's particular strengths is the repeated monitoring of the same area, enabling the authors to evaluate the temporal evolution of the observed morphologies. While this aspect is mentioned in the introduction, we believe that it deserves greater emphasis and could be framed more clearly as a key contribution of the paper. Demonstrating how this temporal perspective enhances our understanding of propeller-induced seafloor changes would strengthen the manuscript and distinguish it from earlier studies. This would be particularly valuable if it were accompanied by quantitative data or estimates of the erosive capacity of vessels that regularly navigate the area (see the related comment below).

- The manuscript should also be reinforced by providing information on hydraulic to support the interpretation that the observed bedforms were induced by ship propellers. It is well established that ship propellers, particularly those of large passenger vessels such as ferries and cruise ships, can generate intense current jets that are capable of inducing significant seafloor erosion (e.g. Guarnieri et al., 2021). These jets can produce high shear stresses, and when these exceed the critical threshold for sediment mobilisation, resuspension and bedform development may occur.

However, the current version lacks supporting evidence to evaluate whether this mechanism is likely under the local conditions. Therefore, we encourage the authors to incorporate data or references related to ship characteristics, such as vessel size, engine power and propeller type, as well as estimates of jet velocity and induced shear stress. This could be achieved by acquiring new data or referencing relevant hydraulic studies of similar vessel types.

Using this information, the authors could estimate the shear stress generated by ship propellers and compare it to the critical threshold for sediment mobilisation in the study area. Presenting this information in the form of a figure or map identifying areas where propeller-induced stresses are likely to exceed erosion thresholds would greatly enhance the robustness of the interpretation and strengthen the argument that the observed morphologies result from propeller activity.

General comments

1. While the manuscript as a whole presents a convincing argument that ship wakes alter seafloor geomorphology, this is not fully reflected in the abstract. We recommend revising the abstract to emphasise the key findings and their implications, especially the observed morphological changes and their relation to vessel traffic.

2. In the Results section, the authors state that 'the mean seafloor depth within the resurveyed area has changed by 0.13 m, indicating that a small amount of sediment has entered the area' (lines 142–143). While this observation is potentially significant, the reported change in depth is relatively small. We therefore recommend that the authors to justify the accuracy of this measurement in light of the characteristics and resolution of the echosounders used. The multibeam echosounder used in the 2024 survey is described in the 'Materials and Methods' section, but very limited information is provided about the system and resolution used in the 2014 survey. Clarifying whether the two datasets are directly comparable in terms of vertical accuracy and processing methods would strengthen the reliability of this result.

3. Dredging is an important human-driven process that significantly alters the morphology of the seafloor in heavily trafficked waterways. It is commonly used to maintain sufficient depth for large vessels (see, for example, Borland et al., 2022; Borland et al., 2022; van Dijk et al., 2021). However, this process is not currently addressed in the introduction. In particular, it would be helpful to establish whether dredging operations are routinely conducted in Kiel Bay. Could the frequency or timing of dredging activities have influenced some of the observed morphological changes observed between the 2014 and 2024

bathymetric surveys? Including this context would strengthen the interpretation of long-term morphological evolution in the study area. Additionally, we suggest that the authors consider discussing the potential influence of storm events on sediment redistribution. In shallow coastal environments, storm-driven hydrodynamics can substantially impact sediment transport and seafloor morphology. A brief discussion of this aspect, alongside any available data or references, would help to contextualise the relative importance of human-induced versus natural processes in shaping the seabed.

4. One of the key limitations of the current manuscript is the absence of quantitative analysis in the 'Results' section. While the figures are visually informative, the interpretations are largely descriptive and qualitative, which restricts the strength of the conclusions that can be drawn. We therefore encourage the authors to incorporate more numerical data to support their observations. Please refer to the 'Specific Comments' for further information and suggestions. For instance, in lines 193–194, the authors suggest that the presence of heavily eroded till beneath the main shipping route (Figure 4) indicates that ship wakes are the primary cause of erosion. However, this figure lacks a quantitative relationship between erosion and shipping activity. A more convincing approach would be to compare the shear stress generated by ship propellers and compare it to the critical threshold for sediment mobilisation in the study area, or generate a cross-sectional plot of the waterway combining bathymetry, rugosity and ship traffic intensity.

5. A more detailed characterisation of maritime traffic, particularly with regard to vessel types, sizes, and frequency, is currently lacking and would greatly enhance the manuscript. Including summary statistics of vessel traffic (e.g., the number and types of vessels per day or their average size) would provide context for the potential morphological impacts discussed in the manuscript. In the current version, the authors use AIS data from vessels longer than 4.5 m travelling at speeds in excess of 2 knots to estimate the intensity of shipping traffic (Figure 4), computing the number of AIS positioning points per 25×25 m grid cell (i.e. density points). While this provides an approximation, the method may lead to biased result, since AIS transmission frequencies vary considerably depending on the type and status of the vessel (ranging from every few seconds to every few minutes). This could result in certain vessels being overrepresented and counted twice within grid cells. This may explain why ship traffic intensity is expressed qualitatively (low–high) in Figure 4, except in panel 4D (number of ferry crossings). Based on the current approach, this may not accurately reflect true crossing frequency. To improve this analysis, we recommend first linking sequential AIS points to create continuous vessel tracks (i.e., individual crossings), and then calculating the number of distinct crossings per grid cell. This would provide a more accurate representation of traffic intensity.

6. The authors attribute changes in seafloor morphology to the wakes generated by shipping traffic. To support this interpretation, they have monitored the wakes of five vessels. However, only one example is presented in detail (Figure 5). As only five vessels were monitored and the authors state that the wakes reached the seafloor in three of those cases, we strongly encourage the inclusion of water column disturbance data for all five vessels. Presenting these results in a figure with multiple subpanels (e.g., one per ship) would not be overly complex, and would allow for a more comprehensive and transparent evaluation of the evidence. It would also be valuable to explore whether there is any relationship between vessel characteristics (such as size or engine power) and the intensity or vertical extent of the wake. This could help to assess whether larger vessels are more likely to induce disturbances that reach the seafloor, and ultimately strengthen the proposed causal link between ship traffic and the observed morphological changes.

7. The manuscript reports on sediment sampling conducted during the 2024 survey. However, it seems that no sediment data was collected in 2014. This raises the question of whether the sediment characteristics observed in 2024 can be considered representative of both time periods. Are there any previously published or archived datasets of bottom sediment samples from the study area that could provide historical context or allow comparison? Given the observed morphological changes between 2014 and 2024, it would be interesting to discuss how these changes may have affected the distribution and grain size of surface sediments over time. We also recommend that the authors explain why they selected the sampling locations. Are these stations representative of the broader morphological features or disturbance gradients observed in the study area? Including such a discussion would help to assess the extent to which the sediment data support broader interpretations of anthropogenic impacts.

8. Building on the earlier suggestion regarding the estimation of propeller-induced shear stress, we encourage the authors to use the available sediment data to calculate the critical shear stress required to mobilise surface sediments in different areas of the study site. They could then use this information to generate an approximate map of critical shear stress values across the study area, which could be compared with the estimated shear stress or bottom current velocities induced by the ship wakes, particularly those of the five monitored vessels. This analysis would allow the authors to determine whether the hydrodynamic energy generated by the wakes is sufficient to cause sediment resuspension and erosion.

9. In lines 232–238, the authors describe the erosion and local redistribution of sediments, forming sorted bedforms and linear ridges. Have the volumes of eroded sediment been estimated, and compared with the volumes accumulated in these depositional features? Including such a comparison would strengthen the interpretation of sediment redistribution and mass balance in the area.

10. In lines 239–250, the authors attempt to extrapolate their findings to the entire Baltic Sea. They do this by assuming that the impact of ships on the seafloor occurs in all shallow areas (less than 20 metres deep) with at least one crossing by a vessel per day (i.e. over 365 crossings per year). The intensity and nature of ship-induced impacts likely depend on various factors, including water depth, traffic frequency, vessel characteristics (e.g., engine power or propeller type) and sediment type, as well as local hydrodynamic conditions. As these parameters can vary considerably across the Baltic Sea, the effect of ship wakes may not be uniform across all shallow regions. While we understand the basis of this assumption, we consider this extrapolation to be somewhat speculative and unnecessary.

11. In lines 254–269 of the Discussion section, the authors address the potential biogeochemical implications of disturbances caused by ships. It is important to note that the study site consists of permeable sandy sediment, allowing strong fluxes of pore waters to across the system, which speeds the cycling of nutrients and carbon in the seafloor (e.g., Rao et al., 2007). This differs markedly from finer-grained sediments, where biogeochemical responses to disturbance may be stronger or more prolonged. In this regard, we note that the study by Tiano et al. (2024), which is cited in the manuscript, was conducted on a finer sediment type. This makes direct comparison with the present study site less applicable. Furthermore, modelling studies by van de Velde et al. (2025) and Zhang et al. (2024), both of which are cited in the manuscript, suggest

that the biogeochemical impacts of trawling in sandy sediments are relatively limited. While we appreciate the authors' attempt to link physical disturbances to biogeochemical processes, we recommend revising this section to emphasise the need for future studies to better constrain the nature and extent of such effects in sandy environments.

12. The order and quality of the figures could be improved to better align with the structure and logical flow of the manuscript. For example, in the introduction (lines 60–62), the authors describe the study area as an important waterway with high vessel traffic. However, this information is not visually represented until Figure 4. It would be more effective to present this figure earlier, ideally when traffic intensity is first discussed, to help reader understand the relevance of the study site. Additionally, the sequence and depth of figure explanations are inconsistent. For example, the manuscript provides a detailed discussion of Figure 2 (lines 124–136) after already explaining Figure 3 (lines 110–121). While Figure 2 is briefly referenced in lines 107–109, this initial mention does not sufficiently support the claims in the text, and the in-depth explanation of the depressions comes later. Reorganising the order of the figures, or at least adjusting how and when they are introduced in the text, would improve the clarity and cohesion of the narrative.

Specific and minor comments:

Line 40: “with centimeter accuracies”, be more specific

Line 42: Please specify book chapter and author(s), not just the full book.

Line 48: “Marine trade is projected to quadruple”, what value are you referring to? Numbers of vessels, hours at sea?

Line 63: “The latter is due to the adjacent entrance of the Kiel Canal, which is among the most heavily used artificial waterways on the globe (considering passages per day)” How many passages per day?

Line 68: “Varying grain size, ranging from clay to boulders”, if this information is available, provide it in map, in the Supplementary if needed.

Line 78: Change “geologic data locate in the central” to “geological data are located in the central”

Line 91: Is Unit 1 or Unit 2?

Line 92: “at least on meter scales”, be more specific and provide ranges of values.

Line 92: “hosts and overall low roughness”, be more specific and provide values

Line 99: “Unit 2 crops out”, what depth changes are these crop outs?

Line 104: “significantly” is used to denote statistical significance. A simple statistical analysis could actually be done with the data. In fact, a boxplot of the roughness in the NE-SW ridge would be very beneficial

Lines 125-126: How many depressions have been mapped? It would strengthen the manuscript to include key quantitative information on the eroded features, such as the number of depressions, and basic morphometric statistics (e.g., minimum, maximum, and mean incision, diameter and individual area).

Lines 142-143: Mean seafloor depth within the re-surveyed area has changed by 0.13 m, indicating that a small amount of sediment has entered the area. Considering the vertical resolution and accuracy of the multibeam echosounder systems and the processing methods used in the 2014 and 2024 surveys, is this value realistic?

Line 189: Please clarify what is meant by “coming from above.” Also, consider whether it is appropriate to directly compare ship wakes with such different processes as marine mammal movements and bottom trawling, given their distinct mechanisms and scales of impact.

Lines 201-202: “Lab experiments have shown that scour holes become wider and longer with increasing rotational speed of ship propellers (Penna et al., 2019)”. It would be helpful to include specific values or ranges from these experiments. Could the results be related or compared to the characteristics of the vessels operating in Kiel Bay?

Line 230: The authors mention the presence of clay, but none of the sediment samples had clay material.

Line 238: Same comment for lines 125-126. Include some information about the number and individual characteristics of the observed depressions.

Lines 243-244: The authors estimate that 7.5% of the Baltic Sea may be affected by sediment alteration induced by ships, based on the 2022 HELCOM AIS shipping density map (HELCOM, 2023). How does this estimate compare with the 6.91% reported by Diaz-Mendoza et al. (2023), which refers to mapped propeller-induced bedforms and scour features, primarily on sandy substrates? It would be interesting to include a discussion about potential versus observed impact metrics.

Lines 245-246: “We note that a single ship crossing will not lead to meter-scale till erosion” I agree, but please, include further information. Is there experimental or observational evidence in the literature that describes the erosive capacity of a single ship passage?

Line 324-326: Where is the data from the listed ferries shown?

Lines 257-259: Please include the reference supporting this statement.

Lines 264-266: Same comment, Include the reference supporting this statement.

Lines 282-286. Could you provide further information about the MBES system used, as well as the data processing workflow? Additional details regarding the resolution and vertical accuracy of the bathymetric data would help assess the reliability of the morphological interpretations.

Figures

Figure 1. Consider enlarging the location map to better visualise the study area in relation to previous work. It would also be useful to include information on maritime traffic density in this figure, as this would provide context for the selection of the study site and its exposure to shipping activity. Additionally, enhancing the relief or vertical exaggeration could improve the visibility of key seafloor morphologies.

Figure 2. As this journal's format is to describe the methods at the end of the journal, it would be helpful if each figure caption describes the resolution of the bathymetry.

Why does panel 2A show the 2024 bathymetry while panel 2B shows data from 2012?

Figure 2B: Why isn't the bathymetry shown on this map? This would provide more context for the figure.

It would also be interesting to include a plot showing the aspect (i.e., the direction of the steepest slope) in relation to the direction of maritime traffic. This would help visually support the idea that the orientation of certain morphological features is influenced by vessel movement.

Figure 3: The reader would find it easier to follow the description of panels B and C if the 2014 and 2024 bathymetries, as well as the differences between them, were annotated. Additionally, the descriptions provided in lines 107-116 should be annotated in this figure.

Figure 4: The y-axis of the inset in panel B is missing its axis label. Please refer to the general comments on improving shipping traffic intensity estimations. Revising the method could enable the authors to transition from a qualitative scale to a more robust, quantitative representation, which would significantly enhance the figure's interpretability and usefulness.

Figure 5. Please consider including additional panels showing the ship wake-induced water column disturbances for all five monitored vessels.

Figure 6. The current figure size is quite small, making it difficult to appreciate key details. We suggest increasing the size of the figure to improve readability. Additionally, please clarify whether panel B is based on hydraulic modelling or inferred directly from the observed morphologies. If it is based on previous hydraulic studies or conceptual models, a reference should be provided.

Reviewer #4

(Remarks to the Author)

Version 1:

Reviewer comments:

Reviewer #1

(Remarks to the Author)

Geersen et al. have clearly reflected on the comments of all four reviewers and extensively revised the article. I am glad that more has been done concerning shear stresses generated by the vessels. The threshold of 4 N/m² found in the laboratory experiment was not exceeded by the vessels according to the model predictions in Figure 7. That is not so surprising to me as a previous postdoc (Dina Vachtman) working with me had found a similar apparent discrepancy in studying streams around the edge of the Dead Sea for her PhD. She found that in-situ water shear stresses calculated from velocimeter data were much smaller than undrained shear strength from vane measurements (I'm sorry I don't have a reference). Apparently, short-lived bursts of turbulence were sufficient to cause stream-bed erosion (in that case, the bed was soft sediment with no bedload if I remember correctly). Perhaps moving particles are important for causing abrasion as the authors suggest, though the process might also simply involve ephemeral bursts of water movement that the averages in Figure 7 don't capture.

Reviewer #2

(Remarks to the Author)

REVIEW:

the authors have responded constructively to most of my comments and suggestions (as well as the other 3 reviewers) in their revised version and I support the progression of this manuscript to publication.

Reviewer #3

(Remarks to the Author)

Dear Authors and Editor,

In this new version, Geersen et al. have addressed the majority of our comments, which we believe have substantially improved the quality and rigor of their findings. In particular, the manuscript provides clearer context regarding previous work on anthropogenic activities modifying the seafloor, and highlight the differences of their work in relation to these prior studies. However, there remain several specific points that we believe the authors should address to further strengthen and support their work, which we outline below:

Major comments

1. As per our comment 3 from our previous review, the authors have addressed the possible influence of waves and bottom

currents, but not the influence of dredging, which is a common practice along heavily trafficked waterways. As mentioned in our previous comment, it would be helpful to contextualize the dredging practices in this area and argue how these could have affected bedforms along the waterways.

2. According to Figure 7, which estimates bed shear stress induced by 5 vessels, Timca generates significantly higher bed shear stress than the other vessels. However, according to the water column disturbance registered by an EK60 echosounder, the strongest disturbance were registered for Annalisa P and Stena Scandinavica, both of which generate a maximum shear stress $< 1 \text{ N/m}^2$. Can the authors justify these differences? Is there a relationship between vessel characteristics and the intensity and/or vertical extent of the wake?

3. In our comment 9 of the previous review (Reviewer #3), we asked whether the authors had calculated the volumes of eroded and re-distributed sediments in order to estimate the net effect of sediment redistribution caused by ship wakes. The authors argue that such a mass balance is not suitable because sediment can enter or leave the study area. However, we believe that, even acknowledging that the system is not fully closed, such a calculation would still provide a valuable and informative approximation of the volume of sediment being remobilized by these activities.

This is particularly relevant because the authors subsequently upscale their results to the entire Baltic Sea and derive an estimate of the total eroded volume (lines 267-284 of the revised manuscript). If such a large-scale estimate is considered meaningful, it would also be informative to calculate the net remobilized volume within the study area itself and compare both values. In fact, we strongly believe that this calculation would provide a clearer insight and reduces many of the assumptions necessary for the extrapolation presented in the manuscript.

Minor comments:

1. The authors have improved their quantification of number of vessel crossings by estimating first individual vessel tracks, and then aggregating these lines, which is now reflected in Figures 1B and 5C, but this method is still not applied for Figure 5A. Please revise this.

2. The CTD casts in Figure 6B were conducted on the same day as the EK60 echosounder data was collected, albeit in different areas. The exact location of the CTDs is given in Fig. 2A, but it would be informative to show in Figure 6 which CTD aligns to which echosounder data. This would also help to identify if there are any relationships between the structure of the water column and the depth of the ship wake recorded by the echosounder.

Thank you very much.

Kind regards

Reviewer #4

(Remarks to the Author)

Version 2:

Reviewer comments:

Reviewer #3

(Remarks to the Author)

We consider the manuscript suitable for publication.

Our only remaining comment is minor. The authors justify in their response why dredging can be excluded, but this clarification does not appear in the manuscript itself. Including a brief statement in the text would improve transparency.

Reviewer #4

(Remarks to the Author)

Note to reviewers

We thank all four reviewers for their thoughtful reviews and valuable comments. We agree with the suggestions and carefully modified text and figures accordingly. Please see our replies in blue font below for details. All references to line numbers below refer to the revised version of the manuscript without track changes.

Reviewer #1

(1) Geersen et al. describe the results of two surveys of the seabed in the Baltic Sea. The data reveal interesting bedforms, including scoured depressions and sand dune trains. Comparison with shipping position data strongly suggests that shipping induced water movements are the cause. One water-column sonar profile supports this interpretation.

We initially decided to show only 1 out of the 5 cases in order to streamline the article and focus on the main line of argumentation. However, we agree that the water column sonar profiles make a strong case for the study. We are therefore happy to add the other 4 profiles as subfigures to Figure 6. Showing all five profiles in the main article also matches well with the newly added shear stress calculations (new Fig. 7) for these 5 vessels.

To prove that the wake seen in the EK60 data breaks through a well stratified water column, we also added CTD profiles to figure 6.

(2) I enjoyed reading the article and think it is publishable after some revision. However, I wondered if the authors have picked the right audience for this. The data do not reveal the biological and geochemical effects that they claim (rather, they are inferred). However, I wondered if there could be an important statement made about how glacial tills erode. To explain this, please look at Shaw et al. (2012). The Canadians collected some very nice data around the Bay of Fundy, where the extreme tides lead to extreme currents. Their data revealed scours where tidal currents are elevated around passages, reaching 10s of m in relative depth. Notably, the margins of the scours are sharp, not blurred.

We are convinced that the impacts of the propeller jets are manifold and relevant for many biologic and geologic processes on a Baltic wide scale. This includes benthic and planktonic ecology, benthic biogeochemistry, physical oceanography, greenhouse-gas fluxes across the sediment-water and air-sea interfaces and many more. We therefore feel that discussing our investigations in a cross-disciplinary journal may motivate other disciplines to investigate a process that has hitherto been largely overlooked.

For the erosion of glacial till, we agree that our data holds information that would be worth to discuss. We now compare our case to the suggested studies from the Bay of Fundy, but also calculated the induced shear stress and compare the values to the (sparse) literature data that are available for glacial till.

New / modified text (line 245): *With more than 16 m, the Bay of Fundy, Canada, host the highest tidal range on the planet. Here, Wu et al. and Shaw et al. showed⁴²⁻⁴³, that glacial till is eroded in areas where near seabed flow velocities exceed 3.5 m/s. With flume experiments, Mier and Garcia further showed⁴⁴ that glacial till is eroded above a shear stress of 4.2 N/m². The calculated flow velocities (Tab. 1) and shear stresses for the selected ships are lower than these thresholds. Nevertheless, the ship-induced shear*

stresses are obviously high enough to cause bedload transport of sand (Fig. 7) and initiate abrasion of glacial till. A similar process is already known from abrasion platforms in shallow water environments in the southwestern Baltic Sea where it occurs due to natural forcings^{32,45}. For the area of our study, model results show that the naturally occurring combined wave-induced and current-induced shear stresses are, however, far too low to initiate motion of sands⁴⁶⁻⁴⁷ and remain far below the levels induced by ship propellers.

(3) Erosion of a sediment might be expected to vary a lot spatially if the sediment has a wide range of grain sizes or if the maximum current at any location has varied a lot over time. Hence, sandy areas tend not to have such abrupt scour margins. If the till is clay-rich, presumably there is a shear-stress threshold above which particles can be extracted by the current. If impacts of saltating particles are important for loosening the other seabed particles, there might also be threshold at which saltation starts, depending on the grain size. Whatever the cause, the scours strongly suggest a threshold shear stress is present. From Shaw et al.'s Figure 15, it should be possible to evaluate the stresses involved (it would involve seabed roughness, which is poorly known, though could be estimated from the seabed photos). Their scours are much deeper and we'd expect till to become stiffer with compaction (Skempton). But I wondered how their stresses might be compared with those in your area. From the internal wave suggested in Figure 5, it might be possible to work out water velocity.

Regarding the water velocities from the sonar data: This is certainly on our list. Instead of using 2D echosounder profiles (e.g. figure 6) we aim to collect high-resolution water-column images across propeller jets with our portable multibeam echosounder. With this data, we will hopefully be able to trace the jets in 3D and transfer into shear stresses. This is, however, beyond the scope of this work and will require careful survey planning, instrument testing, development of new processing workflows and adaptation of particle tracing techniques.

Alternatively, a jet from the propeller might generate higher velocity at the seabed, though I'd be surprised if it were as large as those tidal currents from the Bay of Fundy. Hence, the Bay of Fundy results provide a maximum. There should be geotechnical measurements of in situ stress also to compare against.

We agree that the shear stress generated by ship propellers is critical and should be included in the manuscript. This has been suggested by all 4 reviewers.

To account for this, we calculated propeller-induced seabed shear stress for the investigated vessels based on an empirical model to provide the calculations which was recently used by Krämer et al. (2025) to discuss the formation of detailed bed features in the study area. The calculations were conducted for a water depth of 13 m perpendicular to the sailing direction of the 5 ships, that we traced with the fishery echosounder. In the new figure 7, it is now shown together with the critical shear stresses that are required for the initiation of bedload transport and suspension of medium sand (273.86 μm). The new results show that the calculated shear stress matches the observed morphological pattern generated in the sand grain size range.

We also realize that we synonymously used the terms "seabed erosion" and "glacial till erosion", and that this has led to some confusion. To avoid this, we stick to the term "seafloor

erosion”, which encompasses erosion of the till as well as erosion and redistribution of shallow sands.

Regarding the erosion of glacial till: laboratory data on glacial till erosion is sparse. Results from Mier & Garcia (2011) suggest that the ship-induced shear stress remains below the threshold of erosion for glacial till, but this does not consider the concept of abrasion. We have included the concept of abrasion in the discussion. This process requires bedload transport but not necessarily saltation.

New / modified text (line 196): The calculated seabed shear stresses induced by the five investigated ships (Fig. 7), however, leave little doubt that the energy released by ship wakes is sufficient to affect both the water column and the seabed. In the water column, the wake can break through a well-established summer stratification and mix previously separated water masses. At the seabed, the calculated shear stresses exceed the critical thresholds for sediment motion and are capable of initiating bed-load transport and suspension of sand. In the study area these grain sizes have most likely been eroded from the glacial till by waves^{20,30}, possibly at lower water levels during the postglacial sea level rise. While now out of reach for mobilization by waves and in the absence of strong currents, they are mobilized by ship-induced shear stress. Where the mobile sand layer is thin enough, this may lead to ongoing abrasion of the glacial till.

(4) For the reader's understanding before continuing to the data, I'd suggest some background text on the tidal and wind-blown currents and the waves. Waves should mostly not be important, though extreme waves from an easterly wind may conceivably be large enough to affect the bed in 10-20 m of water. Is the combined stress from these with the propeller effects important?

We have included references showing that the combined wave and current induced shear stress from multi-annual time series is far below the critical thresholds of mobilization of the respective sediment. The propeller-induced stress is orders of magnitude larger and clearly dominant.

New text (line 251): A similar process is already known from abrasion platforms in shallow water environments in the southwestern Baltic Sea where it occurs due to natural forcings^{32,45}. For the area of our study, model results show that the naturally occurring combined wave-induced and current-induced shear stresses are, however, far too low to initiate motion of sands⁴⁶⁻⁴⁷ and remain far below the levels induced by ship propellers.

(5) Lines 20-24. In my opinion, including sentences like these on potential future work leaves a poor impression (it comes across as you seeking funding), while the reader is then distracted from your results. If you remove these sentences, the abstract will be much sharper and leave room for other aspects of your study. Think how the last impression of your abstract helps to motivate a reader to cite its findings.

We agree and removed these two sentences from the abstract.

(6) At risk of promoting myself (hence this is not a request to cite), the authors might look at Mitchell et al. (2021) - your results suggest how the stratigraphy from human activity will develop, hence perhaps a link between your results and the Anthropocene debates. In Mitchell (2014) and Mitchell et al. (2012), we looked into whether or how flows (turbidity currents and tidal currents) lead to erosion. Some of it may be useful for the calculations. I was still not sure if the underlying till (pre-erosion) is expected to be clay rich or not and how cohesive. Could some more background on it be included?

We carefully considered a link to the Anthropocene debate. It would have been tempting to refer to the stratigraphy from human activity and whether it may develop into a new geologic facies. However, ship wake is only one out of many processes (trawling, anchoring, constructions, dumping, etc...) how humans modify the seafloor. We feel that linking anthropogenic seafloor alteration and the Anthropocene debate should be handled carefully in a more holistic review paper on *The Baltic Seafloor in the Anthropocene*.

Detailed suggestions:

Line#

18-19 ...stressor are unquantified, though will include modifications to planktonic life, benthic habitats,... → Changed as suggested

28 Unclear what type of catchment area. I suggest not using this as it can be confused with rainfall catchment. → Changed to "catchment basin"

32 the regions ... -> 56% of the region is shallower than 20 m. (if correct? "below 20 m" is ambiguous) → Changed to "shallower than" (throughout the article)

39 Marie Tharp's article is very broad, though isn't there a publication on Kiel's historical hydrographic surveying? That's more relevant here. → we agree that this would fit better. However, the entire section has been removed during rewriting of the discussion.

45 "in action"? It is not possible to measure during erosion. I'd leave this out.

forces -> influences → both removed during rewriting of introduction

60. This case study area has bathymetry shallower than 20 m and is an area of intense commercial shipping activity due to the entrance to the Kiel Canal, one of (reference if there is one?). → Changed as suggested

67 delete "which represents the landform" → Changed as suggested

71 under -> by → Changed as suggested

78. Readers in my experience seem to like a traditional structure, so you might put this first paragraph in a section marked "Methods and data". → unchanged due to journal format

81 allow us to → Changed as suggested

It would be useful to know a bit about how the data were collected and processed. For example, were corrections made for survey vessel draft (varies with use of marine fuel) and tidal heights? Can you estimate how accurate your values were, as this affects how well changes can be detected? → All information in the method section (journal format requirements)

You might mention the survey direction and the artifacts, which are presumably due to small biases in roll corrections mainly. Hughes Clark et al. might be cited for along-track artifacts. Although the method might not be applicable if your survey lines are not coincident, see Schmitt et al. - sorry for the self-promotion, though it may be useful for some ideas.

Is there a web link for the AIS data? → Yes, in the method section (journal format requirements)

88 Please specify the system used and its acoustic frequency. → information on frequency (400 kHz) and footprint added to the method section

For some general thoughts on how the magnitude of acoustic backscattering might vary and what might cause backscattering (if fish or particles or bubbles caused by propeller cavitation, or other water fluctuations in density/velocity), see Clay and Medwin and Stanton and Clay. There will likely be more recent references. → Thanks for pointing us towards these valuable publications

90 and was subdivided into → Changed as suggested

91 locates -> is → Changed as suggested

92 ...scales, hence it has low roughness (F...). → Changed to "The seafloor within unit 1 appears flat and featureless and thus hosts an overall low roughness (Fig. 1b)."

94 that is elevated → Changed as suggested

95 The boundary between unit 1 and the other units is gradual. → Changed as suggested

99 Unit 2 occurs along → Changed as suggested

100 between 100 and 12 m water depths. → Changed as suggested (10 and 12)

101 On a regional scale -> Regionally → Changed as suggested

103 intersected -> crossed → Changed as suggested

southwest-striking band → Changed as suggested

Please mention the roughness value (make your text more quantitative). → Unchanged, as we feel that this is beyond the scope of our paper. We use the roughness to locate areas that are influenced by erosion from ship propellers. However, we do not draw any conclusions from the degree of roughness (e.g. other processes etc ...). Therefore, we decided to stick to the qualitative description.

104-106 The rough ... - this is a bit vague, I'd suggest deleting this sentence as it is not needed. → Changed as suggested

110 trains of sand dunes with crestlines striking NW-SE. → Changed as suggested

112 that elevate -> elevated → Changed as suggested

114 ... and lie oblique to survey directions, hence are not are a result of survey artifacts. → Changed as suggested

116 It might be better to refer to only Figure 3b and 3c, and mark the smooth patches. Do you mean the areas marked "smooth seafloor"? It is not obvious what you mean by the sand

dunes developing from them -I suggest leaving out this comment (how do you know?). →
Changed as suggested

117 Rather than asserting that unit 3 comprises sand, talk through the grain size measurements first, then say why you think the area more broadly comprises sand (what evidence? - is this marked on hydrographic charts?). → Changed to: "The grain size measurements conducted on six grab samples that were taken at different positions within unit 3 (AL619-55 – AL619-60, Fig. 4a), show that the very shallow sedimentary strata of unit 3 consists of sand."

119. Here you are putting interpretation before observation. Talk through the distribution first then what you infer from it. → Changed as suggested (see above comment)

122 compare? Do you mean based on or similar to? → removed "compare" as not needed

123. I don't know from this text what you have done to get these volumes exactly, consequently it is difficult to interpret the values. I suggest including a short section on the method used within your Methods section. → We think that this was covered sufficiently in the method section.

Unchanged text (line 348): All depressions were filled up to their respective pour points. The newly created grid was then subtracted from the original grid giving the depths of the depressions. Areas that had changed by more than 1 cm were classified and outlined with polygons. The total volume of the depressions as well as the mean change in the study area (red outline, Fig. 2b) were calculated using the GIS Zonal statistics tool.

124 "sums up to" -> is → Changed as suggested

125, which represents 0.355... over the whole area of Unit 3 → Changed as suggested

(if correct- it is unclear what area your 7.2 km² represents, is it the whole of Unit 3 or only the depressions?). → changed to: "we semi-automatically mapped the outlines of the depressions in a 7.2 km² large area in Unit 3 (red outline in Fig. 2b) based on the 2014 bathymetric data"

126 shape -> in plan-view → Changed as suggested

126 striking NE-SW. → Changed as suggested

127 have asymmetric profiles, ... → Changed as suggested

129 remains stable -> is uniform → Changed as suggested

130 The orientations ... varies across the study area.

(if correct - unclear) → Changed as suggested

131 ...the steepest flank.. → Changed as suggested

132 This tendency reverses. There, the steep... → Changed as suggested

138 Delete "wide" → Changed as suggested

139 concentrates on -> overlies → Changed as suggested

141 The 2024 data reveal NE-SW-trending ridges that were not so prominent in the ... →
Changed as suggested

143. 0.13 m value is missing uncertainties. → removed in the revised version following major comment 2 from reviewer 3/4

148-152. Delete this paragraph here, as you need the reader to have studied the AIS data first. Instead this should go in your discussion section. → we see this point, but decided to leave it here for clarification as the text on the AIS data is following directly behind

149 concentrated → Changed as suggested

154 concentrates -> is concentrated → Changed as suggested

156 hardly -> rarely crossed by ships. → Changed as suggested

157 Between 1.6.2024 and 24.9.2024, 1313 ships returned their locations through the AIS in the study area. (please make the text easier to follow) → Changed to: *“Over the 116 days between 1.6 – 24.9.2024, 1,313 ships with a draught >4.5 m and a sailing speed of >2 kn returned their locations through the AIS in the study area”*

159 delete "therefore" → Changed as suggested

161 ... 5648, representing 49 passages per day. → Changed as suggested

163 The maximum draft (10 m) corresponds nearly with the maximum... (9.5 m). Only four vessels with drafts >10 m sailed through the area, likely corresponding to berthings at the Port of Kiel, which currently ... → changed to: *“The maximum draught (10 m) corresponds nearly with the maximum draught allowed in the Kiel Canal (9.5 m). Only four vessels with drafts >10 m sailed through the area, likely corresponding to berthings at the Port of Kiel, which currently accommodates ships with a maximum draught of up to 11.5 m.”*

167 Water column acoustic backscattering following ship passage → Changed as suggested

168-170. the capabilities of the sonar should be in methods. Use this section purely for observations. → moved to method section

172 were conducted in areas with 12-16 m water depths. → Changed as suggested

174 please delete "we clearly" Talk through the observations first, then say how you interpret them. → Changed as suggested

175 Figure 5, for example, shows.... → sentence has been removed as we now show all 5 echograms

180 locates -> lies → Changed as suggested

182 Sentence can be deleted. → Changed to: *“Crossing between 196 - 480 m behind the passing ships, we resolved the wake of all ships in the water column. The sonar volume backscattering strength of the air-bubble-water mixture within the wake reaches values between -70 and -40 dB re 1m-1. For 3 out of the 5 cases, the echogram shows a propeller wake reaching to the sea surface as well as down to the seafloor (Fig. 6a).”*

184. Reader cannot evaluate this. Is it possible to include the other profiles as an electronic supplement? → added as subfigure to Fig. 6

185 "The intensity ..." - I'm not sure this is true. If you know the wavelength and amplitude of the waveform, I think you can estimate a current speed. For surface waves for comparison this is possible (Masselink and Hughes). → agree, added: "two-dimensional" for explanation

189 I'm not so sure it would be similar to bottom trawling. A better analogy might be simply the effects of surface waves in shallow water. → agree, added: "wave forcing in shallow waters (Schwarzer et al., 2010)" as additional comparison.

210 by the two corridors of erosion underlying the shipping lanes (Fig. 6). → Changed as suggested

224 sedimentation -> sedimentary particles → Changed as suggested

226 There has been quite a lot of work done by the group at the UK's IOS, including Neil Kenyon, Arthur Stride, Bob Belderson and others. It is fair to say it was largely descriptive (non-quantitative). → we acknowledge this information. I checked some of the many manuscripts but did not find something that seemed directly comparable in terms of volumes.

234. The Berne et al. citation is misleading. I think you need to have outlined the tidal currents, wind-blown currents and surface waves as background or part of introduction → agree, added: "*A similar process is already known from abrasion platforms in shallow water environments in the southwestern Baltic Sea where it occurs due to natural forcings^{32,45}. For the area of our study, model results show that the naturally occurring combined wave-induced and current-induced shear stresses are, however, far too low to initiate motion of sands⁴⁶⁻⁴⁷ and remain far below the levels induced by ship propellers.*" for explanation.

236 The depth changes in the sands here, of up to, outpace ... → Changed as suggested

338 This statistic has been mentioned before, so rephrase this. → combined with latter sentence on upscaled erosion in the Baltic Sea

246 ... 1895 and since then hundreds as well as to other parts of the Baltic Sea. → Changed as suggested

248 Delete sentence (Many of them..) unless you have some values on these drafts. E.g., were there ships, such as battleships, of sufficient size to merit mentioning? → deleted

257 Towards greater depths, ... → Changed as suggested

281. I guess the format of the journal leads to the methods being here. In my opinion, this leads to confusion as the reader has to flit back and forth to understand the study in detail. I would have submitted this work to a journal with a more traditional format. → please see our reply to comment 2 (this reviewer).

285 delete "provided in xyz format and"

using the GMT software modules → Changed as suggested

291 used for correcting the soundings for refraction. → Changed as suggested

292 spline is an interpolation - not sure what you mean. → it is the name of a filter algorithm in the software

296 You need to explain what UTM means. → We think that this is basic knowledge that can be looked up if the reader is unfamiliar with it.

How were the tidal height data recorded (tide gauge, if so where, or model?). → we recorded height via the RTK data and then reduced to mean sea level using the German Combined Quasigeoid (GCG2016).

303 depths of the depressions. → Changed as suggested

You need to explain also how the roughness in Figure 1B was calculated (roughness depends on the spatial scale of measurement and the mean surface that it is measured against). → Regarding the magnitude of roughness, we feel that this is beyond the scope of our paper. We use the roughness to locate areas that are influenced by erosion from ship propellers. However, we do not draw any conclusions from the degree of roughness (e.g. other processes etc ...). Therefore, we decided to stick to the qualitative description.

319 Were there any vessels with drafts <4.5 m travelling at exceptional speed? → Yes, certainly many small vessels going faster than 2 kn.

332-3 This kind of information is needed to interpret the figures shown, so it is strange having it here. Hence, in my opinion, a more traditional format is better. → agree, but see above

335. It would have been nice to know a bit more about these samples. E.g., was the rest of the sediment similar in terms of grain size? What were the maximum particle sizes present? Were any clays present? → More information on the regional grain size distribution is given by Krämer et al. (2025). We now refer to this article in our introduction and discussion.

439-440. I suggest putting the things in brackets at the end of the sentence to make it more easily read. Hence: "The traffic Canal (green and brown arrows, respectively)." → Changed as suggested

442. You are mixing up observations and interpretation here. I would simply highlight the high roughness coinciding with sea lines. → Changed as suggested

442 "Flat and featureless unit 1..." The reader cannot see unit 1. Again, I would simply highlight that this area has low relief or leave this out (obvious). → deleted

447. Somewhere you might mention the survey orientation and that the NW-SE lines are artifacts of the data (for people without our backgrounds). → added text: "*Note that the survey was oriented in northeast – southwest direction and that the lines in the inset figures are artefacts of the data.*"

456 Differential -> Bathymetric → Changed as suggested

459. It might be good to clarify what the phi scale means (for non-geologists). Values in mm might also be useful (second horizontal axis). → we feel that with the "silt" and "sand" labels in this figure it should be understandable

460 Selection of data showing how two areas changed bathymetry between 2014 and 2024. → Changed as suggested

463. Please mention in the caption what these intensity values mean (i.e., simply give min/max with units for B and C). → added to new Figure 1

Figure 4C - difficult to work out the water depth from the colour table shown. I would guess water depth is around 15 m? → we redid all bathymetric figures and now stick to one color scale with variable min/max values, so that the morphologic details are more pronounced.

Figure 5 - The undulating reflections/backscattering layers are not pycnocline exactly. Rather they are due to fish and other causes of acoustic impedance variations within the water, which are being moved up and down by an internal wave acting on the pycnocline. → we agree, they are an indication for the presence of a pycnocline. We added the CTD profiles to

figure 6 to show that there are indeed pycnoclines (i.e. that the entire water column is highly stratified). However, for the scope of this work we leave the labeling unchanged in order to avoid confusion.

Figure 6 in B, you mention disturbed thermocline, though here it should be pycnocline. → changed as suggested

474 Figure caption does not make sense. B does not show sediment biogeochemistry. → changed to: “shallow sediment composition”.

I enjoyed reading the article and look forward to reading it in print.

Neil Mitchell, University of Manchester, May 2025.

Clay, C.S. and Medwin, H., 1977. *Acoustical oceanography: principles and applications*. Wiley Interscience, New York, 544 pp.

Hughes Clarke, J.E., 2003. Dynamic motion residuals in swath sonar data: Ironing out the creases. *Int. Hydrog. Rev.*, 4: 6-23.

Masselink, G. and Hughes, M.G., 2003. *Introduction to coastal processes and geomorphology*. Arnold, London, 354 pp.

Mitchell, N.C., Huthnance, J.M., Schmitt, T. and Todd, B., 2012. Threshold of erosion of submarine bedrock landscapes by tidal currents. *Earth surface processes and landforms*, 38: 627-639.

Mitchell, N.C., 2014. Bedrock erosion by sedimentary flows in submarine canyons. *Geosphere*, 10: doi:10.1130/GES01008.01001.

Mitchell, N.C., Jerrett, R. and Langman, R., 2022. Dynamics and stratigraphy of a tidal sand ridge in the Bristol Channel (Nash Sands banner bank) from repeated high-resolution multibeam echo-sounder surveys. *Sedimentology*, 69: 1051-1082.

Schmitt, T., Mitchell, N.C. and Ramsay, A.T.S., 2008. Characterizing uncertainties for quantifying bathymetry change between time-separated multibeam echo-sounder surveys. *Cont. Shelf Res.*, 28: 1166-1176.

Shaw, J., Todd, B.J., Li, M.Z. and Wu, Y., 2012. Anatomy of the tidal scour system at Minas Passage, Bay of Fundy, Canada. *Mar. Geol.*, 323-325: 123-134.

Skempton, A.W., 1970. The consolidation of clays by gravitational compaction. *Q. J. Geol. Soc. London*, 125: 373-411.

Stanton, T.K. and Clay, C.S., 1986. Sonar echo statistics as a remote-sensing tool: volume and seafloor. *IEEE J. Oceanic Eng.*, OE-11(1): 79-96.

Reviewer #2

This research article presents a really interesting and new perspective on ship wake erosion. The combination of repeat multibeam surveys, geologic sampling, water column imagery and morphological analyses provides compelling evidence for the erosion and redistribution of sediment and hard substrate, in areas with heavy vessel traffic. There are some general and specific comments I have made below, which I hope are helpful to the authors in revising this manuscript. I feel there are some things that would benefit from revision, but they are mostly related to manuscript structure rather than methods. There are some sentences that were a bit confusing as a reader, which I have highlighted for revision in the specific comments section.

General comments:

(1) The introduction is a bit confusing to me. It provides quite a lot of detail about the Baltic, which is fine. But I feel it could be improved to start with an introduction about shipping impacts, and then move into the Baltic as a case study. I think the introduction needs to provide more about the state of the literature on shipping wake impacts. As a read There is a lot of detail about the Baltic formation and geology, but it lacks information about the shipping context up front, for example where is the Kiel Canal?

We agree that shipping impacts should be mentioned as the general topic of the manuscript, before moving to the Baltic Sea as a case study. We therefore restructured the introduction and added some further references on ship wake as an anthropogenic stressor (Erbe et al., 2020; Nylund et al., 2025; Krämer et al., 2025). Please see the revised text with track changes for details on how we restructured the discussion.

Yes, Kiel Canal is central, and we apologize that it was not shown in the initial submission. It has now been added to Figure 1.

(2) The figures are generally well compiled and clear. But there are a few suggestions in the specific comments that would help those not as familiar with the Baltic geography to understand spatial interactions, or moving figures forward or back.

We agree. Please see the revised figures and our comments below for details.

(3) I think the change detection work is great. But please make sure you aren't providing more accuracy than you are able to with these systems? You mention you can see less than 20cm (line 136) but then report things to centimetre difference.

I think there's been a misunderstanding between resolution and horizontal/vertical accuracy. The 20 cm refer to the sonar beam footprint. We rephrased the section to avoid confusion.

Revised text (line 133): *Depressions without visible boulders may result from the limited horizontal resolution due to the sonar beam footprint which hinders detection of objects smaller than 20 cm at the given water depth.*

Specific comments:

Line 31 – “In the southwestern Baltic Sea (west of 15°E), the regions below 20 m even account for 56% of the entire area.” This is a confusing sentence? Perhaps revise? → changed to: “... shallower than 20 m ...”

Line 36 – “The first bathymetric measurements with an echosounder, interestingly already conducted in the Bay of Kiel, took place over 100 years ago.” Is this relevant to this paper? Perhaps more focus could be placed on shipping impacts over time rather than mapping the Baltic? → agree, we significantly rewrote the introduction to account more for shipping impacts.

Line 48 – this reads more like an introductory sentence to me, the paper is about shipping impacts, with a demonstrated case study in the Baltic. Perhaps this is the way to structure the introduction? → agree, see above

Line 50 – one thing that has been quite extensively studied for a long time is ship noise, perhaps worth mentioning that in this review section? I realise this isnt about noise, but surely this would also be an impact here. → agree, added to the introduction “*Increased marine traffic over the last decades has already led to an increased number of bigger ships, more powerful propulsions systems, and increased ship noise*⁶.”

Figure 1 – it feels like this figure is missing information about the shipping intensity. I see it is shown in Fig 4A, is it possible to bring this into Fig 1 to clearly show why the Baltic is a good place to study ship traffic impacts? → agree, moved information on shipping intensity from fig. 4 to the new fig. 1.

Also, can you add a colour scale onto the figure so we know what the inset depth range is? Or is it the same as 1A? → added to the new figure 1a

Fig 1B – I like that you show the roughness, but its not clear why this is needed in Fig 1? It seems to be more of a morphological analysis and could be placed later. That way you could bring shipping intensity into this figure and make the context inset bigger and with more labels. → We moved the roughness map to Fig. 2

Paragraph 59-76 – the introduction about the Baltic generally, from paragraph 1 could fit nicely into this paragraph. → agree, we significantly rewrote the introduction to account more for shipping impacts.

Line 61 – can you label the Kiel canal on Fig 1? Those unfamiliar with this region may not know where this is. → The canal has been added to Figure 1

Line 71 – is the Bay of Kiel and Kiel Bay the same place? → Yes, and to avoid confusion, we only use “Bay of Kiel” in the revised version of the manuscript

Line 76 – any sort of survey in particular? → changed to “repeated bathymetric surveys”

Line 76 – extra “,” that isnt necessary → Changed as suggested

Line 78 – This sentence is missing something? “All hydroacoustic and geologic data locate in the central Bay of Kiel” → no, fine from our side

Line 82/282 – were the 2014 and 2024 data collected with the same or similar systems? And what frequencies were these systems using? This could be important when doing change detection as if you have a vastly different frequency then the changes could be due to that

(somewhat) as well as erosive mechanisms → information on frequency (where possible) has been added to the method section

Line 87 – which type of echosounder was used for this? → we used an EK60 fishery split-beam echosounder. This is specified in the method section

Figure 2 – the depth colour scales appears to change from 2A to the insets, it might be clearer if colour scales are kept the same? → we redid all bathymetric figures and now stick to one color scale with variable min/max values, so that the morphologic details are more pronounced.

Fig 2(B) - the label says 2012 bathymetry, which is this? You have not described use of a 2012 dataset? Is this a typo? → Yes, thanks for spotting this. It was a typo and has been changed to 2014.

Fig 2C – this graph seems to show that both the blue and red depressions have slope aspects the vary from SW to NE. I see there is a peak where higher slope blue depressions are more to the NE and red depressions with a higher slope are more SW facing. But perhaps a statistical summary of this would be helpful to show that this difference can be quantified? And/or a density plot showing the distribution of points? To help with more clearly articulating this point. Its quite hard to see what you describe and so its not fully convincing to me just yet. → We fully understand this concern. However, instead of adding more statistics at this point, we decided to refer to the study of Krämer et al. (2025), which was published in *Geomorphology* in the meantime. The authors discuss the directionality of these features in detail. We believe that it is sufficient for this study to highlight the main trend between SW and NE facing depressions.

New text (line 56): Krämer et al. further discussed¹⁵ the small-scale morphological characteristics of these features and showed that their genesis is controlled by changes in bottom shear stress induced by the propeller wakes from passing ships.

Line 89 – I think unit should be singular, or you could say description of morphological units. → Changed as suggested

Line 108 – is there any seafloor backscatter to show the substrate variation and if these “stones” are relatively hard or different substrate compared to surrounds? → unfortunately, no backscatter is available for this study

Line 124- “Their cumulative volume sums up to 2,556,000 m³, which accounts to 0,355 m³/m².” this sentence is unclear to me. I don’t understand how we got to a volume over an area, please explain how you determined the volume of the pockmarks (was is individually determined or their area multiplied by the average depth?) and help me understand why this is relevant? → agree that this needs to be explained in more detail.

New text (line 121): Using a morphological workflow optimized for mapping of seabed depressions²⁹, we semi-automatically mapped the outlines of the depressions in a 7.2 km² large area in Unit 3 (red outline in Fig. 2b) based on the 2014 bathymetric data. Their cumulative volume is 2,556,000 m³, which represents 0,355 m³/m² over the whole investigated area.

Line 134- you mention the limits of detection here but in the scale bar in Fig 3a, you have the differencing down to 2 decimal places. Perhaps its worth only having one decimal place in

the figure given the detection limit? Similarly you report mean depth changes of 0.13 m in line 143, should this be just 1 decimal place? → agree, changed to 1 decimal place.

For the mean depth change, we agree that it is relatively small. Because our study area is not confined like a port, sediment can be transported in and out of it. This further complicates the comparison of mean seafloor depth over the 10 years. Referring to the mean change also somehow obscures the up to 1.5 m of vertical change that we observe in some places.

For all the reasons above, we decided to remove this aspect from the main text. This also helps to emphasize the strength of the repeated monitoring (also see or reply to “major weakness A” from reviewer 3/4).

Line 137 – perhaps this title should be “repeat” instead of time-lapse? To me time-lapse signals many instances, whereas repeat is resurveying the area at a different time-step. Its finicky I know but perhaps more accurate. → Agree and changed to “Repeat bathymetric survey”

Figure 3 – to my eyes the 2014 and 2024 datasets appear to have different resolutions, with the 2024 looking to be higher resolution than 2014 – which makes sense given the respective ages of each. Perhaps in this figure caption or by labelling, its worth mentioning these figures show different resolution data but that you did the differencing with the same resolution? → Yes, the 2024 data was regridded to 1 m. This, however, was already mentioned in the method section.

Unchanged text (line: 344): The data sets were resampled to 1 m and then subtracted using the raster calculator to create the differential bathymetry maps.

Line 154 – perhaps “lane” is a better word than “stripe”? → Agree and changed to “lanes”

Line 157 – can you also provide the duration on this timeframe? → agree, changed to “Over the 116 days between 1.6 – 24.9.2024”

Line 157 – can you rephrase this to say “ 1313 individual ships were identified using AIS, with a draft of...” → agree, changed to: “*Over the 116 days between 1.6 – 24.9.2024, 1,313 ships with a draught >4.5 m and a sailing speed of >2 kn returned their locations through the AIS in the study area*”

Line 158 – can you quantify the max number of times a single ship passed in 1 day → we could but maybe not necessary here. It is n=2.

Line 160 – does the AIS data have information about vessel tonnage? → no reliable information provided

Line 162 – what does sub-weekly mean? → more than once a week, it changes, so no precise information available here (but maybe also not needed)

Line 160-166 – this is a bit confusing, maybe I misunderstood but you say that the max draft is 9.5 then say 4 ships came in with 10 m draft because the max draft is 11.5m. I think maybe because im not familiar with where the canal is relative to the port and so could this be more clearly explained and perhaps add these locations onto figures so people can follow the geography? → location of Kiel Canal added to figure 1.

We also changed the text to: “The maximum draught (10 m) corresponds nearly with the maximum draught allowed in the Kiel Canal (9.5 m). Only four vessels with drafts >10 m

sailed through the area, likely corresponding to berthings at the Port of Kiel, which currently accommodates ships with a maximum draught of up to 11.5 m.”

Line 168 – change “towards the seafloor” to “in the water column” → *Changed as suggested*

Line 169 – this sentence has already been mentioned above I think “It is optimized for water column analyses and capable of tracing the spatial extent and duration of wake induced water column disturbance” – and its probably more suited to methods too? → *agree, we removed the sentence*

Line 171 – how far (in meters or distance measurement) behind the ships were you approximately? → *Added text (line 170): Crossing between 196 - 480 m behind the passing ships, we resolved the wake of all ships in the water column.*

Line 180 – change to “...consists of till but is located outside the main shipping corridor..” → *changed to: Unit 2, which lies outside of the main shipping corridor,*

Line 181 – the word “major” seems a bit ambiguous as to what you mean. Does that mean there was some erosion but not much? What is it from? I suggest changing “major” to “propeller-induced” or similar → *agree, changed to propeller-induced*

Line 185 – just out of interest, any idea how long after the ship has gone the water remains disturbed? → *I would speculate that it persists for some tens of minutes. But we need to find out in the future.*

Line 202-204 – I agree but I feel like if you mention this it would be good to mention what the oceanic conditions are in the region? → *We added Gräwe and Burchard (2012) as a reference for the oceanic conditions.*

Line 205 – minor grammar, should read “they ARE locateD within the busiest section” → *Changed as suggested*

Fig 6 – I really like that the authors have brought everything together but im not sure what panel A adds to this figure? I think B and C provide the process you are trying to describe well. Perhaps you could just reference your earlier figure 4 or 2 in the caption instead of repeating it in this one? → *We agree and removed panel (a) from the entire figure.*

Line 210 – do you know how many of the ships would have two propellers? → *no, unfortunately also a bit off topic for us geologists*

Line 214 – it would be great if this was included in Fig 6, how the ship directionality impacts the turbulence and sediment redistribution. “Together, the two propellers therefore displace the sediment to a central line underneath the hull of the ship, which leads to the formation of the linear ridges.” → *this is sketched in figure 6b (new figure 8b).*

Line 232 – are these “sorted bedforms” the ridges? → *no, the dunes. We now refer to Krämer et al. (2025) for more details. New text (line 260): The sandy fraction is redeposited within the area, partly forming sorted bedforms¹⁵.*

Line 243 – I agree but wouldn't deeper waters also have bigger ships? → *yes, but only to a draft of 15.4 meters (Baltimax) which is the maximum for entering the Baltic Sea.*

Line 248/9 – perhaps change this to “to cause turbulent flow down to at least 20 m water depth” to simplify? → *sentence has been removed in response to comments from reviewers 1 and 3.*

Line 270-272 – its not clear if this is from your study or from the literature generally? Please specify and add references as needed → *Agree, changed to: Our study suggests that ship wakes have the potential to erode*

Line 277-279 – is there more we should investigate in these substrates too? Perhaps adding a suggestion to look at infaunal community disturbance or noise pollution or similar would be a nice end suggestion for future research direction? → *yes, but here we wanted to give a practical recommendation for hydrographic agencies. That is why we specifically refer to the depth change.*

Line 291 – were you able to do an SVP in the wake at any point? → *no, unfortunately not. We are aiming for it. But station work is usually not allowed directly within the shipping lanes..*

Line 304 – don't you say somewhere else that you can only see changes >20 cm? → *yes, but this only refers to the modified grid that is artificially produced during the morphological analysis.*

Line 475 – does this figure show sediment biogeochemistry, or ecosystem function? I realise these are impacts of the processes you describe in the paper, but this figure isnt really showing that, its showing the physical process of wake erosion only (in water and sediments). Perhaps refine the caption to reflect what the figure shows? → *agree, changed to: "Induced alterations include variations of seafloor morphology, water column integrity, and shallow sediment composition with potential implications for ecosystem functioning and element budgets on a Baltic scale."*

We leave the "potential implications" to refer to ecosystem functioning

Reviewer #3 (Remarks to the Author):

Summary

The manuscript “Ship wake induced water column mixing and meter-scale seabed erosion in the Baltic Sea” by Geersen et al (Ref.NCOMMS-25-33870) provides new insights into the impact of commercial shipping on the morphology of the seafloor in shallow-water environments, particularly in areas with heavy marine traffic. We find the topic and the idea of the manuscript of scientific interest. However, we believe that additional evidence and further development are needed to reinforce several key aspects of the analysis and interpretation, and to strengthen the manuscript’s conclusions. We therefore encourage the authors to consider the following recommendations and revise the manuscript accordingly.

This review report discusses two key issues relating to the interpretation of erosive seafloor features. It offers suggestions, asks questions and highlights textual corrections to improve the clarity, coherence and robustness of the manuscript.

We thank both reviewers for the very careful review of our initial submission. We have incorporated most suggestions.

Major weakness

(A) The authors provide a detailed characterisation of the erosional bedforms generated by repeated ship propeller passage in Kiel Bay. These features are of great interest as they provide valuable insights into the impact of human activity on seafloor morphology and sedimentary cover. However, the manuscript would benefit from clearer explanations of its novelty and of how it builds upon previous work in the region.

For the first time, we show repeated bathymetric surveys from a main shipping route in the Baltic Sea. The new data demonstrate that erosion and redistribution of sedimentary strata can alter absolute water depth by more than 10% on annual to decadal timescales. These impacts are profound, with direct consequences for benthic ecosystems, biogeochemical cycling, and maritime safety, and they demonstrate the urgent need for customized monitoring across the Baltic Sea. We have now strengthened the discussion to emphasize this better.

Furthermore, by directly imaging water-column disturbances behind passing ships, we demonstrate the disruption and breakdown of the pycnocline using a split-beam echosounder. This provides unique observational evidence and the mechanism of ship-induced internal mixing processes not previously documented. To strengthen this point, we have now incorporated additional wake surveys from five vessels of different sizes and have rewritten the abstract and manuscript to better emphasize this novelty.

However, we acknowledge that our initial references to the recently published studies of Díaz-Mendoza et al. (2025) and Krämer et al. (2025) did not sufficiently explain how our work advances beyond theirs. To clarify this, we have revised the introduction as follows:

New text (line 54): In a recent publication, Díaz-Mendoza et al. report¹⁸ for the first time, that the Bay of Kiel hosts various bedforms that may relate to propeller-induced scouring caused by ships. They documented subaqueous dunes and scouring features in water

depths between 10-19 m. Krämer et al. further discussed¹⁵ the small-scale morphological characteristics of these features and showed that their genesis is controlled by changes in bottom shear stress induced by the propeller wakes from passing ships. We now demonstrate that ship wakes not only generate distinct morphological features at the seafloor, but also completely alter the internal structure of the water column by mixing stratified waters of different oxygen, temperature, and salinity content. With repeated bathymetric surveys we further show that absolute changes in water depth caused by the erosion and redistribution of sedimentary strata locally exceed 10% on annual to decadal timescales. The implications of these findings for benthic ecosystems on a Baltic-wide scale require dedicated monitoring strategies

Specifically, similar bedforms have recently been described in the south-western Baltic Sea (Díaz-Mendoza et al., 2023; 2025). Díaz-Mendoza et al. suggest that propeller-induced features, such as small subaqueous dunes, scour pits and comet marks, were found to affect 6.91% of the impacted seabed, particularly over sandy substrates with boulders (see Figure 5 and Table 3 in Díaz-Mendoza et al., 2025). However, the introduction by Geersen et al. only briefly mentions these studies, without clearly explaining how the present work builds upon, complements, or differs from them.

We agree, please see our answer above on how we rewrote the introduction.

Regarding the 6.91% of the impacted seabed observed by Díaz-Mendoza et al., 2025): we agree that this comparison is valuable, and therefore added the following text to the discussion:

New text (line 272): The upscaled magnitude of alteration on a Baltic scale, ranges at a similar level compared to the 6.91% reported¹⁸ by Díaz-Mendoza et al., who documented mapped propeller-induced bedforms and scour features, primarily on sandy substrates

Therefore, a more thorough discussion in the introduction of existing evidence of similar morphologies in this or other regions, and of the specific new insights provided by the present study would help to situate this work more clearly within the broader scientific context, clarifying its contribution to advancing knowledge in this field.

We realize that the reference to the recently published manuscripts (Díaz-Mendoza et al., 2025; Krämer et al., 2025) was not sufficient to clearly explain how our work builds upon these investigations. To account for this, we added the following text to the introduction:

New text (line 54): In a recent publication, Díaz-Mendoza et al. report¹⁸ for the first time, that the Bay of Kiel hosts various bedforms that may relate to propeller-induced scouring caused by ships. They documented subaqueous dunes and scouring features in water depths between 10-19 m. Krämer et al. further discussed¹⁵ the small-scale morphological characteristics of these features and showed that their genesis is controlled by changes in bottom shear stress induced by the propeller wakes from passing ships.

One of the study's particular strengths is the repeated monitoring of the same area, enabling the authors to evaluate the temporal evolution of the observed morphologies. While this aspect is mentioned in the introduction, we believe that it deserves greater emphasis and

could be framed more clearly as a key contribution of the paper. Demonstrating how this temporal perspective enhances our understanding of propeller-induced seafloor changes would strengthen the manuscript and distinguish it from earlier studies. This would be particularly valuable if it were accompanied by quantitative data or estimates of the erosive capacity of vessels that regularly navigate the area (see the related comment below).

We agree that it should be clearer how our study distinguishes from the recent work of Díaz-Mendoza et al. in Kiel Bay and that we should refer in more detail to other studies on related topics elsewhere. We therefore added the following text to the introduction:

Modified text (line 38): The morphological impact of ships anchoring on the seafloor has been studied by different authors⁷⁻⁸. Ship bubble wakes and their detection near the sea surface by optical and acoustic sensing has also been widely investigated⁹⁻¹⁰. Nylund et al. suggested¹¹ that passing ships can trigger methane emissions from natural sources in coastal areas. For confined areas like ports, Guarnieri et al. discussed¹² the influence of ship propellers on sediment erosion and accumulation. What has, however, only been marginally researched is the impact of the wake of passing ships on the water column, sedimentation patterns, seafloor morphology and the benthic ecosystem in open waters¹³⁻¹⁵. For the Venice lagoon, Madricardo et al. and Scarpa et al. showed¹⁶⁻¹⁷ that depression wakes created by large ships cause the shoreline to retreat at different locations thereby threatening the stability of anthropogenic structures.

Modified text (line 54): In a recent publication, Díaz-Mendoza et al. report¹⁸ for the first time, that the Bay of Kiel hosts various bedforms that may relate to propeller-induced scouring caused by ships. They documented subaqueous dunes and scouring features in water depths between 10-19 m. Krämer et al. further discussed¹⁵ the small-scale morphological characteristics of these features and showed that their genesis is controlled by changes in bottom shear stress induced by the propeller wakes from passing ships. We now demonstrate that ship wakes not only generate distinct morphological features at the seafloor, but also completely alter the internal structure of the water column by mixing stratified waters of different oxygen, temperature, and salinity content. With repeated bathymetric surveys we further show that absolute changes in water depth caused by the erosion and redistribution of sedimentary strata locally exceed 10% on annual to decadal timescales.

(B) The manuscript should also be reinforced by providing information on hydraulic to support the interpretation that the observed bedforms were induced by ship propellers. It is well established that ship propellers, particularly those of large passenger vessels such as ferries and cruise ships, can generate intense current jets that are capable of inducing significant seafloor erosion (e.g. Guarnieri et al., 2021). These jets can produce high shear stresses, and when these exceed the critical threshold for sediment mobilisation, resuspension and bedform development may occur.

We are thankful for pointing us towards the work of Guarnieri et al. who looked at the process in confined areas like ports. We added the reference and now also mention that we investigate open waters. This also helps to explain how our work distinguishes from earlier studies (see comment A from this reviewer above).

Modified text (line 42): For confined areas like ports, Guarnieri et al. discussed¹² the influence of ship propellers on sediment erosion and accumulation. What has, however,

*only been marginally researched is the impact of the wake of passing ships on the water column, sedimentation patterns, seafloor morphology and the benthic ecosystem in open waters*¹³⁻¹⁵

However, the current version lacks supporting evidence to evaluate whether this mechanism is likely under the local conditions. Therefore, we encourage the authors to incorporate data or references related to ship characteristics, such as vessel size, engine power and propeller type, as well as estimates of jet velocity and induced shear stress. This could be achieved by acquiring new data or referencing relevant hydraulic studies of similar vessel types.

We agree that the information on the characteristics of the different ships is crucial and therefore added table 1 to the main manuscript.

Using this information, the authors could estimate the shear stress generated by ship propellers and compare it to the critical threshold for sediment mobilisation in the study area. Presenting this information in the form of a figure or map identifying areas where propeller-induced stresses are likely to exceed erosion thresholds would greatly enhance the robustness of the interpretation and strengthen the argument that the observed morphologies result from propeller activity.

We agree that the shear stress generated by ship propellers is critical and should be included in the manuscript. This has been suggested by all 4 reviewers.

To account for this, we calculated propeller-induced seabed shear stress for the investigated vessels based on an empirical model to provide the calculations which was recently used by Krämer et al. (2025) to discuss the formation of detailed bed features in the study area. The calculations were conducted for a water depth of 13 m perpendicular to the sailing direction of the 5 ships, that we traced with the fishery echosounder. In the new figure 7, it is now shown together with the critical shear stresses that are required for the initiation of bedload transport and suspension of medium sand (273.86 μm). The new results show that the calculated shear stress matches the observed morphological pattern generated in the sand grain size range.

General comments

(1) While the manuscript as a whole presents a convincing argument that ship wakes alter seafloor geomorphology, this is not fully reflected in the abstract. We recommend revising the abstract to emphasise the key findings and their implications, especially the observed morphological changes and their relation to vessel traffic.

We rewrote the abstract (also considering comment 5 from reviewer #1) in order to emphasise the key findings and their implications.

New / modified text (line 16): *We unveil substantial seafloor erosion, including up to 1.5 m variation in water depths, over 10 years that clearly relates to vessel traffic. By imaging water column disturbance behind passing ships, we trace wake turbulence to the seafloor and show the breakdown of a strongly stratified water column and a possible excitement of internal waves, likely increasing the mixing of oxygen, nutrients, and greenhouse*

gases. While the environmental consequences of this anthropogenic stressor are unquantified, our findings leave little doubt that they include modifications to marine ecosystems and element budgets on a Baltic-wide scale.

(2) In the Results section, the authors state that 'the mean seafloor depth within the resurveyed area has changed by 0.13 m, indicating that a small amount of sediment has entered the area' (lines 142–143). While this observation is potentially significant, the reported change in depth is relatively small. We therefore recommend that the authors to justify the accuracy of this measurement in light of the characteristics and resolution of the echosounders used. The multibeam echosounder used in the 2024 survey is described in the 'Materials and Methods' section, but very limited information is provided about the system and resolution used in the 2014 survey. Clarifying whether the two datasets are directly comparable in terms of vertical accuracy and processing methods would strengthen the reliability of this result.

We agree that the mean absolute change in depth is relatively small. Because our study area is not confined like a port, sediment can be transported in and out of it. This further complicates the comparison of mean seafloor depth over the 10 years. Referring to the mean change also somehow obscures the up to 1.5 m of vertical change that we observe in some places.

For all the reasons above, we decided to remove this aspect from the main text. This also helps to emphasize the strength of the repeated monitoring (also see or reply to “major weakness A” from this reviewer above).

(3) Dredging is an important human-driven process that significantly alters the morphology of the seafloor in heavily trafficked waterways. It is commonly used to maintain sufficient depth for large vessels (see, for example, Borland et al., 2022; Borland et al., 2022; van Dijk et al., 2021). However, this process is not currently addressed in the introduction. In particular, it would be helpful to establish whether dredging operations are routinely conducted in Kiel Bay. Could the frequency or timing of dredging activities have influenced some of the observed morphological changes observed between the 2014 and 2024 bathymetric surveys? Including this context would strengthen the interpretation of long-term morphological evolution in the study area. Additionally, we suggest that the authors consider discussing the potential influence of storm events on sediment redistribution. In shallow coastal environments, storm-driven hydrodynamics can substantially impact sediment transport and seafloor morphology. A brief discussion of this aspect, alongside any available data or references, would help to contextualise the relative importance of human-induced versus natural processes in shaping the seabed.

We have included references showing that the combined wave and current induced shear stress from multi-annual time series is far below the critical thresholds of mobilization of the respective sediment. The propeller-induced stress is orders of magnitude larger and clearly dominant.

New text (line 251): *A similar process is already known from abrasion platforms in shallow water environments in the southwestern Baltic Sea where it occurs due to natural forcings^{32,45}. For the area of our study, model results show that the naturally occurring*

combined wave-induced and current-induced shear stresses are, however, far too low to initiate motion of sands⁴⁶⁻⁴⁷ and remain far below the levels induced by ship propellers.

(4) One of the key limitations of the current manuscript is the absence of quantitative analysis in the 'Results' section. While the figures are visually informative, the interpretations are largely descriptive and qualitative, which restricts the strength of the conclusions that can be drawn. We therefore encourage the authors to incorporate more numerical data to support their observations. Please refer to the 'Specific Comments' for further information and suggestions. For instance, in lines 193–194, the authors suggest that the presence of heavily eroded till beneath the main shipping route (Figure 4) indicates that ship wakes are the primary cause of erosion. However, this figure lacks a quantitative relationship between erosion and shipping activity. A more convincing approach would be to compare the shear stress generated by ship propellers and compare it to the critical threshold for sediment mobilisation in the study area, or generate a cross-sectional plot of the waterway combining bathymetry, rugosity and ship traffic intensity.

We follow the advice and calculate the shear stress generated by the propellers of the 5 ships that we traced with the split-beam echosounder and compare it to the critical threshold for sediment mobilisation in the study area. Please also see the new table 1 and the new figure 7 for details.

(5) A more detailed characterisation of maritime traffic, particularly with regard to vessel types, sizes, and frequency, is currently lacking and would greatly enhance the manuscript. Including summary statistics of vessel traffic (e.g., the number and types of vessels per day or their average size) would provide context for the potential morphological impacts discussed in the manuscript. In the current version, the authors use AIS data from vessels longer than 4.5 m travelling at speeds in excess of 2 knots to estimate the intensity of shipping traffic (Figure 4), computing the number of AIS positioning points per 25 × 25 m grid cell (i.e. density points). While this provides an approximation, the method may lead to biased result, since AIS transmission frequencies vary considerably depending on the type and status of the vessel (ranging from every few seconds to every few minutes). This could result in certain vessels being overrepresented and counted twice within grid cells. This may explain why ship traffic intensity is expressed qualitatively (low–high) in Figure 4, except in panel 4D (number of ferry crossings). Based on the current approach, this may not accurately reflect true crossing frequency. To improve this analysis, we recommend first linking sequential AIS points to create continuous vessel tracks (i.e., individual crossings), and then calculating the number of distinct crossings per grid cell. This would provide a more accurate representation of traffic intensity.

We are thankful for the advice on the varying AIS transmission frequencies. We follow the suggestion and now link AIS points to continuous tracks before counting the number of tracks in one grid cell. We added this information to the method section and also revised figure 5.

New / modified text (line 371): The data transmitted by the ferries COLOR FANTASY, COLOR MAGIC, STENA GERMANICA, STENA SCANDINAVICA, AURA SEAWAYS and VICTORIA SEAWAYS were handled separately in addition (Fig. 5c). For these ships, we first linked all AIS data points of each vessel to a continuous track, before counting the

number of tracks in each grid cell. This allows us to derive the absolute number of crossings for each grid cell (Fig. 5c).

(6) The authors attribute changes in seafloor morphology to the wakes generated by shipping traffic. To support this interpretation, they have monitored the wakes of five vessels. However, only one example is presented in detail (Figure 5). As only five vessels were monitored and the authors state that the wakes reached the seafloor in three of those cases, we strongly encourage the inclusion of water column disturbance data for all five vessels. Presenting these results in a figure with multiple subpanels (e.g., one per ship) would not be overly complex, and would allow for a more comprehensive and transparent evaluation of the evidence. It would also be valuable to explore whether there is any relationship between vessel characteristics (such as size or engine power) and the intensity or vertical extent of the wake. This could help to assess whether larger vessels are more likely to induce disturbances that reach the seafloor, and ultimately strengthen the proposed causal link between ship traffic and the observed morphological changes.

We initially decided to show only 1 out of the 5 cases in order to streamline the article and focus on the main line of argumentation. However, we agree that the water column sonar profiles actually make a strong case for the study. We are therefore happy to add the other 4 profiles as subfigures to Figure 5.

Showing all five profiles in the main article also matches well with the newly added shear stress calculations for these 5 vessels.

(7) The manuscript reports on sediment sampling conducted during the 2024 survey. However, it seems that no sediment data was collected in 2014. This raises the question of whether the sediment characteristics observed in 2024 can be considered representative of both time periods. Are there any previously published or archived datasets of bottom sediment samples from the study area that could provide historical context or allow comparison? Given the observed morphological changes between 2014 and 2024, it would be interesting to discuss how these changes may have affected the distribution and grain size of surface sediments over time. We also recommend that the authors explain why they selected the sampling locations. Are these stations representative of the broader morphological features or disturbance gradients observed in the study area? Including such a discussion would help to assess the extent to which the sediment data support broader interpretations of anthropogenic impacts.

From a geological perspective, the shallow sedimentary strata have not changed over the 10 years. Here, and elsewhere in the Western Baltic Sea, it is mostly composed of glacial till from the Weichselian glaciation that is overlain by Holocene sediments. However, these shallow Holocene sediments – mainly sands in our case - are moved around by the wakes from the ship propellers. The thickness of these sands (and the seafloor morphology) therefore does change over time. However, we do not have any repeated grab samples at the same location, that could be used to derive a time series of sediment composition – comparable to our repeated multibeam survey.

(8) Building on the earlier suggestion regarding the estimation of propeller-induced shear stress, we encourage the authors to use the available sediment data to calculate the critical shear stress required to mobilise surface sediments in different areas of the study site. They could then use this information to generate an approximate map of critical shear stress values across the study area, which could be compared with the estimated shear stress or bottom current velocities induced by the ship wakes, particularly those of the five monitored vessels. This analysis would allow the authors to determine whether the hydrodynamic energy generated by the wakes is sufficient to cause sediment resuspension and erosion.

We fully agree. Please see our response to “major weakness B” and “major comment 4” from this reviewer for details.

(9) In lines 232-238, the authors describe the erosion and local redistribution of sediments, forming sorted bedforms and linear ridges. Have the volumes of eroded sediment been estimated, and compared with the volumes accumulated in these depositional features? Including such a comparison would strengthen the interpretation of sediment redistribution and mass balance in the area.

We realize that we are not looking at a closed (confined) system. Because sediment can enter and leave the study area, we refrain from such mass balance calculations.

(10) In lines 239–250, the authors attempt to extrapolate their findings to the entire Baltic Sea. They do this by assuming that the impact of ships on the seafloor occurs in all shallow areas (less than 20 metres deep) with at least one crossing by a vessel per day (i.e. over 365 crossings per year). The intensity and nature of ship-induced impacts likely depend on various factors, including water depth, traffic frequency, vessel characteristics (e.g., engine power or propeller type) and sediment type, as well as local hydrodynamic conditions. As these parameters can vary considerably across the Baltic Sea, the effect of ship wakes may not be uniform across all shallow regions. While we understand the basis of this assumption, we consider this extrapolation to be somewhat speculative and unnecessary.

We would not call the calculations speculative although we admit that they have large uncertainties. However, we do account for these uncertainties and clearly refer to them in the discussion:

However, we are convinced that the impacts of the propeller jets are manifold and relevant for many biologic and geologic processes on a Baltic wide scale. This includes benthic and planktonic ecology, benthic biogeochemistry, physical oceanography, greenhouse-gas fluxes across the sediment-water and air-sea interfaces and many more. We therefore feel that referring to the absolute volumes on a Baltic wide scale, may motivate other disciplines to investigate a process that has hitherto been largely overlooked.

(11) In lines 254–269 of the Discussion section, the authors address the potential biogeochemical implications of disturbances caused by ships. It is important to note that the study site consists of permeable sandy sediment, allowing strong fluxes of pore waters to across the system, which speeds the cycling of nutrients and carbon in the seafloor (e.g., Rao et al., 2007). This differs markedly from finer-grained sediments, where biogeochemical

responses to disturbance may be stronger or more prolonged. In this regard, we note that the study by Tiano et al. (2024), which is cited in the manuscript, was conducted on a finer sediment type. This makes direct comparison with the present study site less applicable. Furthermore, modelling studies by van de Velde et al. (2025) and Zhang et al. (2024), both of which are cited in the manuscript, suggest that the biogeochemical impacts of trawling in sandy sediments are relatively limited. While we appreciate the authors' attempt to link physical disturbances to biogeochemical processes, we recommend revising this section to emphasise the need for future studies to better constrain the nature and extent of such effects in sandy environments.

We thank the reviewer for this valuable comment and the interesting perspective. In response, we have substantially expanded our discussion of the biogeochemical implications, with particular emphasis on the role of sandy sediments.

New text (line 302): Exchange processes in sandy sediments are driven by pressure gradients induced by overlying currents and waves⁵⁷⁻⁶¹. Ship passages may induce transient pressure gradients that enhance sediment-water exchange, while at the same time sand mobilization could reduce exchange⁶². However, which of the processes dominates remains unexplored. Additionally, in benthic habitats, the physical instability of the sediment surface can disrupt the establishment of structured infaunal and epifaunal communities, favouring opportunistic and disturbance-tolerant species while displacing more sensitive taxa⁶³. These shifts can cascade through sedimentary biogeochemical processes, altering bioirrigation, bioturbation, and benthic-pelagic nutrient fluxes. These disturbances may further impact higher trophic levels through changes in food availability, water clarity, and chemical habitat conditions.

New text (line 315): Sandy sediments, the dominant substrate in most coastal settings, are particularly vulnerable, as their exchange processes are tightly coupled to pressure gradients and flow dynamics. Yet, these habitats and their role in mediating benthic–pelagic exchange remain underrepresented in current research.

(12) The order and quality of the figures could be improved to better align with the structure and logical flow of the manuscript. For example, in the introduction (lines 60–62), the authors describe the study area as an important waterway with high vessel traffic. However, this information is not visually represented until Figure 4. It would be more effective to present this figure earlier, ideally when traffic intensity is first discussed, to help reader understand the relevance of the study site. Additionally, the sequence and depth of figure explanations are inconsistent. For example, the manuscript provides a detailed discussion of Figure 2 (lines 124–136) after already explaining Figure 3 (lines 110–121). While Figure 2 is briefly referenced in lines 107–109, this initial mention does not sufficiently support the claims in the text, and the in-depth explanation of the depressions comes later. Reorganising the order of the figures, or at least adjusting how and when they are introduced in the text, would improve the clarity and cohesion of the narrative.

We agree and reorganized the figures.

As suggested, we now show the high vessel traffic in the study area already in figure 1. We also added figure7 to show the bottom shear stress generated by the ship propellers.

Please see the revised figures and our comments below for details.

Specific and minor comments:

Line 40: “with centimeter accuracies”, be more specific → we agree, however, the sentence has been removed during rewriting of the introduction

Line 42: Please specify book chapter and author(s), not just the full book. → the reference has been removed during rewriting of the introduction

Line 48: “Marine trade is projected to quadruple”, what value are you referring to? Numbers of vessels, hours at sea? → In this case we were referring to demand for port facilities. However, the sentence has been removed during rewriting of the introduction.

Line 63: “The latter is due to the adjacent entrance of the Kiel Canal, which is among the most heavily used artificial waterways on the globe (considering passages per day)” How many passages per day? → 90 per day. Modified text: *This case study area has bathymetry shallower than 20 m and is an area of intense commercial shipping activity due to the entrance to the Kiel Canal (Fig. 1), one of the most heavily used artificial waterways on the globe with around 90 passages per day on average*

Line 68: “Varying grain size, ranging from clay to boulders”, if this information is available, provide it in map, in the Supplementary if needed. → not available in map view

Line 78: Change “geologic data locate in the central” to “geological data are located in the central” → Changed as suggested

Line 91: Is Unit 1 or Unit 2? → Unit 1

Line 92: “at least on meter scales”, be more specific and provide ranges of values. → as the small-scale variations are not visible on the scale of fig. 2, we removed the “at least on meter scale”

Line 92: “hosts and overall low roughness”, be more specific and provide values → Unchanged, as we feel that this is beyond the scope of our paper. We use the roughness to locate areas that are influenced by erosion from ship propellers. However, we do not draw any conclusions from the degree of roughness (e.g. other processes etc ...). Therefore, we decided to stick to the qualitative description.

Line 99: “Unit 2 crops out”, what depth changes are these crop outs? → changed to “Unit 2 occurs along the northwestern and eastern margins of the studied area in water depths between 10 and 14 m (Fig. 1).”

Line 104: “significantly” is used to denote statistical significance. A simple statistical analysis could actually be done with the data. In fact, a boxplot of the roughness in the NE-SW ridge would be very beneficial → we agree and removed “significantly”.

Regarding the magnitude of roughness, we feel that this is beyond the scope of our paper. We use the roughness to locate areas that are influenced by erosion from ship propellers. However, we do not draw any conclusions from the degree of roughness (e.g. other processes etc ...). Therefore, we decided to stick to the qualitative description.

Lines 125-126: How many depressions have been mapped? It would strengthen the manuscript to include key quantitative information on the eroded features, such as the number of depressions, and basic morphometric statistics (e.g., minimum, maximum, and

mean incision, diameter and individual area). → We refrained from providing these specific quantitative numbers as many depressions amalgamate into larger areas preventing a reliable statistical analysis of individual features. Instead, to support our interpretations, we focus on their geometrical characteristics (Lines 126–133) and restrict further quantitative numbers to volume estimates and the endmember values for slope and depth.

Lines 142-143: Mean seafloor depth within the re-surveyed area has changed by 0.13 m, indicating that a small amount of sediment has entered the area. Considering the vertical resolution and accuracy of the multibeam echosounder systems and the processing methods used in the 2014 and 2024 surveys, is this value realistic? → We agree that the mean absolute change in depth is relatively small. Because our study area is not confined like a port, sediment can be transported in and out of it. This further complicates the comparison of mean seafloor depth over the 10 years. Referring to the mean change also somehow obscures the up to 1.5 m of vertical change that we observe in some places.

For all the reasons above, we decided to remove this aspect from the main text. This also helps to emphasize the strength of the repeated monitoring (also see our reply to “major weakness A” from this reviewer above).

Line 189: Please clarify what is meant by “coming from above.” Also, consider whether it is appropriate to directly compare ship wakes with such different processes as marine mammal movements and bottom trawling, given their distinct mechanisms and scales of impact. → From a geological perspective, depressions in the seabed are often assigned to sub-seafloor fluid flow and seepage. There we feel that for geologists it is important to note that the depression creating forcing is coming from above (i.e. from the water column).

Lines 201-202: “Lab experiments have shown that scour holes become wider and longer with increasing rotational speed of ship propellers (Penna et al., 2019)”. It would be helpful to include specific values or ranges from these experiments. Could the results be related or compared to the characteristics of the vessels operating in Kiel Bay? → there is not enough information in the article to upscale the results from the analogue experiments to the Baltic Sea. At least for people like us, who are not based in engineering or marine geotechnics.

Line 230: The authors mention the presence of clay, but none of the sediment samples had clay material. → This is because we only used the upper 2 cm for the grain size analysis. We note that it would have been useful to also analyze the deeper sediments but did not store the grab samples.

Line 238: Same comment for lines 125-126. Include some information about the number and individual characteristics of the observed depressions. → Please see our reply to comment on lines 125-126

Lines 243-244: The authors estimate that 7.5% of the Baltic Sea may be affected by sediment alteration induced by ships, based on the 2022 HELCOM AIS shipping density map (HELCOM, 2023). How does this estimate compare with the 6.91% reported by Díaz-Mendoza et al. (2023), which refers to mapped propeller-induced bedforms and scour features, primarily on sandy substrates? It would be interesting to include a discussion about potential versus observed impact metrics. → we agree and added the following sentence:

New text (line 272): The upscaled magnitude of alteration on a Baltic scale, ranges at a similar level compared to the 6.91% reported¹⁸ by Díaz-Mendoza et al., who documented mapped propeller-induced bedforms and scour features, primarily on sandy substrates

Lines 245-246: “We note that a single ship crossing will not lead to meter-scale till erosion” I agree, but please, include further information. Is there experimental or observational evidence in the literature that describes the erosive capacity of a single ship passage? → not that we are aware of. However, there is information on the erodibility of till. We therefore added the following text to the discussion:

New text (line 245): With more than 16 m, the Bay of Fundy, Canada, host the highest tidal range on the planet. Here, Wu et al. and Shaw et al. showed⁴²⁻⁴³, that glacial till is eroded in areas where near seabed flow velocities exceed 3.5 m/s. With flume experiments, Mier and Garcia further showed⁴⁴ that glacial till is eroded above a shear stress of 4.2 N/m². The calculated flow velocities (Tab. 1) and shear stresses for the selected ships are lower than these thresholds. Nevertheless, the ship-induced shear stresses are obviously high enough to cause bedload transport of sand (Fig. 7) and initiate abrasion of glacial till.

Line 324-326: Where is the data from the listed ferries shown? → it is shown in Fig. 5c. We added the information to the text.

Lines 257-259: Please include the reference supporting this statement. → agree, and added Berg et al. (2022)

Lines 264-266: Same comment, Include the reference supporting this statement. → we added Thrush and Dayton (2002) who looked at this from a bottom trawling perspective.

Lines 282-286. Could you provide further information about the MBES system used, as well as the data processing workflow? Additional details regarding the resolution and vertical accuracy of the bathymetric data would help assess the reliability of the morphological interpretations. → we added information on the footprint and the frequency to the method section

Figures

Figure 1. Consider enlarging the location map to better visualise the study area in relation to previous work. It would also be useful to include information on maritime traffic density in this figure, as this would provide context for the selection of the study site and its exposure to shipping activity. Additionally, enhancing the relief or vertical exaggeration could improve the visibility of key seafloor morphologies. → we agree and split figure 1 into new figures 1 and 2. Information on traffic intensity as well as a larger map of the study area is now shown in figure 1. A larger map of the seafloor morphology is now shown in Figure 2.

Figure 2. A this journal’s format is to describe the methods at the end of the journal, it would be helpful if each figure caption describes the resolution of the bathymetry. → we agree and added the information to figures 2 and 3 (where the datasets are shown for the first time).

Why does panel 2A show the 2024 bathymetry while panel 2B shows data from 2012? → Thanks for pointing this out. It was a typo, and we corrected it to 2014.

Figure 2B: Why isn't the bathymetry shown on this map? This would provide more context for the figure. → we agree, but more color would also make the figure too busy and would make it more difficult to see the red and blue dots.

It would also be interesting to include a plot showing the aspect (i.e., the direction of the steepest slope) in relation to the direction of maritime traffic. This would help visually support the idea that the orientation of certain morphological features is influenced by vessel movement. → in principle yes, but aspects plots are again very busy (and colorful). That is why we decided to conceptualize with just the red and blue dots.

Figure 3: The reader would find it easier to follow the description of panels B and C if the 2014 and 2024 bathymetries, as well as the differences between them, were annotated. Additionally, the descriptions provided in lines 107-116 should be annotated in this figure. → we agree and checked that all details (*single depressions with stone, merged depressions, Sand dunes, smooth seafloor patches, NE-SW ridges*) are labeled in the figures B and C

Figure 4: The y-axis of the inset in panel B is missing its axis label. Please refer to the general comments on improving shipping traffic intensity estimations. Revising the method could enable the authors to transition from a qualitative scale to a more robust, quantitative representation, which would significantly enhancing the figure's interpretability and usefulness. → we agree that this was misleading and changed to %

Figure 5. Please consider including additional panels showing the ship wake-induced water column disturbances for all five monitored vessels. → We initially decided to show only 1 out of the 5 cases in order to streamline the article and focus on the main line of argumentation. However, we agree that the water column sonar profiles make a strong case for the study. We are therefore happy to add the other 4 profiles as subfigures to Figure 5. Showing all five profiles in the main article also matches well with the newly added shear stress calculations for these 5 vessels

Figure 6. The current figure size is quite small, making it difficult to appreciate key details. We suggest increasing the size of the figure to improve readability. Additionally, please clarify whether panel B is based on hydraulic modelling or inferred directly from the observed morphologies. If it is based on previous hydraulic studies or conceptual models, a reference should be provided. → we agree that the figure was too small to see all details. We decided to remove panel (A) differential changes and enlarged panels B and C. It is based on the observations and therefore no reference was added.

Reviewer #4 (Remarks to the Author):

Reviewer #3

In this new version, Geersen et al. have addressed the majority of our comments, which we believe have substantially improved the quality and rigor of their findings. In particular, the manuscript provides clearer context regarding previous work on anthropogenic activities modifying the seafloor, and highlight the differences of their work in relation to these prior studies. However, there remain several specific points that we believe the authors should address to further strengthen and support their work, which we outline below:

Major comments

1 As per our comment 3 from our previous review, the authors have addressed the possible influence of waves and bottom currents, but not the influence of dredging, which is a common practice along heavily trafficked waterways. As mentioned in our previous comment, it would be helpful to contextualize the dredging practices in this area and argue how these could have affected bedforms along the waterways.

According to the *WSV (Wasserstraßen- und Schifffahrtsverwaltung des Bundes)* webpage on dredging activities, Kiel is not mentioned as an area where dredging takes place. Unfortunately, the webpage is only available in German and not really appropriate for citation.

https://www.kuestendaten.de/DE/Services/Nassbaggerarbeiten/Nassbaggerarbeiten_node.html

To confirm this, we contacted the WSA and got a clear response that no maintenance dredging operations were performed in the investigation area (Waterways and Shipping Authority Baltic Sea (WSA), personal communication).

Dredging activity in the Baltic Sea is also covered in the HELCOM HOLAS data. In agreement with the WSA, the HELCOM report shows no dredging activity at the location of our study. We added the explanation and references to the dataset to the discussion.

New text (line 224): *Dredging can also be ruled as genetic origin of the observed morphologies, as no activity is recorded for the area³⁷⁻³⁸.*

37 HELCOM HOLAS II Dataset: Dredging areas 2011-2016 (2018).

38 HELCOM HOLAS 3 Dataset (2023).

Furthermore, the observed patterns do not align with dredging artifacts. Dredging creates broad trenches or disposal mounds. Our data reveal highly localized scour depressions around boulders and prominent linear ridges (elevated seafloor). The formation of the linear ridges is flow-parallel and located precisely in the busiest lanes, aligning with the confluence zone of propeller wash. Furthermore, the reversed asymmetry of the scour depressions (Fig. 3A, C) across the inward/outward corridors provides a decisive, bi-directional flow forcing signature that is uniquely to the high-volume passage of vessels. We maintain that the presented morphological evidence clearly demonstrates that the observed seabed patterns originate from ship wake and propeller jet interactions, not from dredging activities.

2 According to Figure 7, which estimates bed shear stress induced by 5 vessels, Timca generates significantly higher bed shear stress than the other vessels. However, according to the water column disturbance registered by an EK60 echosounder, the strongest disturbance were registered for Annalisa P and Stena Scandinavica, both of which generate a maximum shear stress $< 1 \text{ N/m}^2$. Can the authors justify these differences? Is there a relationship between vessel characteristics and the intensity and/or vertical extent of the wake?

We see the point and realize that we did not mention explicitly enough that the EK60 data were recorded at different distances to the passing ships.

The fact that the strongest wake was observed for the ship with the closest point of approach (Annalisa P) indicates that wake energy decreases with time. To account for this, we calculated the true distance between the ships (accounting for the position of the GPS Antenna) and added information on the closest (Annalisa P) and farthest (Timca) EK60 measurement. We also added a sentence to the discussion that mentions the decrease in wake energy over time.

Modified text:

Line 170: *Crossing between 176 m (Annalisa P) - 416 m (Timca) behind the passing ships, we resolved the wake of all ships in the water column.*

Line 195: *The most intense water column disturbance was documented for the ship with the closest point of approach (Annalisa P), suggesting that the intensity of the wake decreases over the first couple of minutes after a ship has passed.*

Caption Fig. 6: *It was imaged between 176 m (Annalisa P) - 416 m (Timca) behind the passing ships.*

We further want to emphasize, that Figure 7 models the potential for sediment erosion at the seabed, whereas Figure 6A documents the acoustic backscattering strength (Sv) of the air-bubble-water wake mixture. The latter depends on complex, non-linear factors (e.g., bubble size spectra and crossing geometry) that do not scale directly with the maximum modeled near-bed shear stress.

3 In our comment 9 of the previous review (Reviewer #3), we asked whether the authors had calculated the volumes of eroded and re-distributed sediments in order to estimate the net effect of sediment redistribution caused by ship wakes. The authors argue that such a mass balance is not suitable because sediment can enter or leave the study area. However, we believe that, even acknowledging that the system is not fully closed, such a calculation would still provide a valuable and informative approximation of the volume of sediment being remobilized by these activities.

This is particularly relevant because the authors subsequently upscale their results to the entire Baltic Sea and derive an estimate of the total eroded volume (lines 267-284 of the revised manuscript). If such a large-scale estimate is considered meaningful, it would also be informative to calculate the net remobilized volume within the study area itself and compare both values. In fact, we strongly believe that this calculation would provide a clearer insight and reduces many of the assumptions necessary for the extrapolation presented in the manuscript.

There is the fact that we are not looking at a closed system (as mentioned in our previous reply). Furthermore, we have no control whether the changes occur gradually over the 10 years

or repeatedly within individual events. Resolving this question is only possible with regular and event-based seagoing campaigns over at least some years. Without this knowledge, there is the danger that a proposed rate of change (i.e. amount of volume mobilized over 10 years) would be used to upscale the results to industrial timescales, despite the insufficient temporal resolution. To avoid this temporal blur, we stick to the volume of the depressions in a large area (i.e. the 2014 dataset), avoiding working with temporally unconstrained rates of change.

Minor comments:

1. The authors have improved their quantification of number of vessel crossings by estimating first individual vessel tracks, and then aggregating these lines, which is now reflected in Figures 1B and 5C, but this method is still not applied for Figure 5A. Please revise this.

We agree. Changed as suggested. Modified text:

Method section (line 368): *We first linked all AIS data points of each vessel to a continuous track, before counting the number of tracks in 25 * 25 m grid cells. This allows us to derive the absolute number of crossings for each grid cell (Fig. 5).*

Figure 5 caption: *Ship traffic intensity in relation to seafloor morphology (A) Ship traffic intensity in the study are between 1 June - 24 September 2024, considering only ships with a draught >4.5m and a speed >2kn. Shown is the number of crossings for 25*25 m sized grid cells. Inset diagram shows the draught distribution for the considered ships. (B) Detailed bathymetry in the area of the two linear ridges that were observed in the 2024 bathymetric data. (C) Course of the six ferries that link Kiel with Sweden, Norway and Lithuania (same area covered by figure 5B).*

2. The CTD casts in Figure 6B were conducted on the same day as the EK60 echosounder data was collected, albeit in different areas. The exact location of the CTDs is given in Fig. 2A, but it would be informative to show in Figure 6 which CTD aligns to which echosounder data. This would also help to identify if there are any relationships between the structure of the water column and the depth of the ship wake recorded by the echosounder.

The CTDs were conducted at the start and endpoints of the multibeam bathymetric profiles that were collected on the same day and in the same area as the EK60 data. However, they were not collected parallel to the collection of the EK60 data. Therefore, there is no clear relation - neither in time nor in space - between individual casts and individual EK60 profiles. We therefore decided not to link specific CTD casts with specific EK60 profiles in Fig. 6.

The CTD data, however, clearly demonstrates the stratification of the water column (at the day when the EK60 data was recorded) which can be seen in all casts.

Reviewer #3

We consider the manuscript suitable for publication.

Our only remaining comment is minor. The authors justify in their response why dredging can be excluded, but this clarification does not appear in the manuscript itself. Including a brief statement in the text would improve transparency.

Dredging activity in the Baltic Sea is covered in the HELCOM HOLAS data and we decided to cite this data to confirm no dredging activity at the location of our study.

New text: Dredging can also be ruled as genetic origin of the observed morphologies, as no activity is recorded for the area³⁷⁻³⁸.

37 HELCOM HOLAS II Dataset: Dredging areas 2011-2016 (2018).

38 HELCOM HOLAS 3 Dataset (2023).